# A memory-driven auditory program ensures selective and precise vocal imitation in zebra finches

Wan-chun Liu [1✉], Michelle Landstrom [1], Gillian Schutt[1], Mia Inserra[1] & Francesca Fernandez[1]

In the vocal learning model, the juvenile first memorizes a model sound, and the imprinted memory gradually converts into vocal-motor output during the sensorimotor integration. However, early acquired memory may not precisely represent the fine structures of a model sound. How do juveniles ensure precise model imitation? Here we show that juvenile songbirds develop an auditory learning program by actively and attentively engaging with tutor's singing during the sensorimotor phase. The listening/approaching behavior requires previously acquired model memory and the individual variability of approaching behavior correlates with the precision of tutor song imitation. Moreover, it is modulated by dopamine and associated with forebrain regions for sensory processing. Overall, precise vocal learning may involve two steps of auditory processing: a passive imprinting of model memory occurs during the early sensory period; the previously acquired memory then guides an active and selective engagement of the re-exposed model to fine tune model imitation.

[1] Department of Psychological and Brain Sciences, Colgate University, Hamilton, NY, USA. ✉email: wliu1@colgate.edu

Vocal learning in songbirds consists of two learning phases that overlap to some degree[1]. During the early sensory learning phase, juveniles first passively listen to the song of their adult tutor and form an auditory memory. Then, during the ensuing sensorimotor phase, internally encoded auditory memory is gradually transformed into vocal motor output through auditory feedback[1–4] (but see a different view by Roper and Zann[2,5]). It is generally thought that early exposure to adult tutors is sufficient for tutor song acquisition[6,7]. However, the early-encoded model memory may not precisely represent the fine acoustic structures of the complex song repertoire of the tutor song[8,9]. Moreover, the growing number of field studies suggest that, even if model memory is first acquired during the sensory learning phase, later re-exposure to the model during the sensorimotor learning phase is critical to ensure selective attention on final song commitment and precise model imitation during territory establishment[10–12]. Imprecise imitation may impair effective social communication among close neighbors and lessen breeding success[13]. We hypothesize that previously acquired model memory may later guide the juvenile's selective attention and anticipation toward the tutor and serve ecological adaptive functions[14]. Here we show that zebra finches (*Taeneopygia guttata*) universally develop an auditory learning program that allows juveniles to actively and selectively engage in a previously exposed tutor song for refinement of complex repertoire imitation.

## Results

**Characteristics of attentive listening approaching behavior**. Under a semi-natural social setting, we tracked each juvenile zebra finch and its social interaction with an adult tutor during part of the sensitive period of vocal learning, from 30 to 65 days post hatching (dph) (Fig. 1a). At ~ 44–65 dph, juvenile males engaged in a robust, attentive listening and approaching behavior immediately following the tutor singing (Fig. 1b). The juvenile's listening approaching behavior is defined here as immediate orientation and flying in close proximity (< 5 cm of distance) toward its tutor upon hearing the tutor singing (i.e., within 5 s after the onset of the tutor singing), followed by a temporary freeze of ongoing behavior (e.g., feeding and preening) with strict silence for at least 1 s (Fig. 1c, d and Supplementary Movie 1). The juveniles who had a higher attentive approaching rate (i.e., number of approaching movements/ number of tutor songs) were more likely to peck their tutor's beak immediately after approaching the tutor (< 5 s after the tutor song, Pearson's correlation, $R^2 = 0.52$, $n = 26$ birds; Supplementary Fig. 1 and Supplementary Movie 2). Attentive but less-motivated behaviors were frequently observed during this period but were not included as listening approaching behavior. For example, juveniles often stopped ongoing behavior during the tutor singing without executing tutor-approaching behavior. Attentive but less-motivated, non-approaching behavior was also described in a previous study[15]. Additionally, the listening approaching behavior was not included if the distance between the tutor and the approached juvenile was > ~ 5 cm.

Tutor-approaching movement occurred immediately after the onset of the tutor singing (time latency = 2.7 ± 0.6 s, mean ± SEM, after the tutor's production of the first introductory note; $n = 26$ birds; Fig. 1f). The onset of approaching behavior varied greatly among individuals. The juveniles who had a higher approaching rate also approached their tutor faster (i.e., shorter time latency between the onset of tutor song and tutor approaching, Fig. 1f). Remarkably, some of the most attentive, motivated juveniles even initiated their approaching movement milliseconds before the onset of tutor singing (Fig.1f, Supplementary Movie 3). We

speculate this highly attentive behavior is possibly initiated by the juvenile's anticipation of the tutor singing by closely observing the tutor's gesture movement before singing.

The listening approaching behavior occurred universally among juveniles in a short time window during the sensorimotor integration period, starting at around 38–40 dph and peaking around 46–52 dph ($n = 33$ birds; Fig. 1b). This time frame parallels the timing of song circuit formation[16] and the surge of "plastic song" production[17], characterized by the emergence of recognizable syllable structures and sequences with different strategies[4,18]. This attentive listening approaching behavior was associated with male-specific song learning in zebra finches and was seen in the juvenile females to a much lesser extent (Fig. 1e).

**Social influence on listening approaching behavior**. The developmental trajectory of juvenile attentive listening varied among families. Similar-age siblings of the same clutch often developed a similar trajectory of approaching behavior (Supplementary Fig. 2).

To investigate whether tutor singing-induced listening/ approaching behavior is associated with a juvenile's overall approaching toward its tutors under all social contexts (tutor singing or not), we tracked the approaching movement from a juvenile to its tutor under all social conditions during the sensitive period of 30–65 dph ($n = 17$ birds). The overall approaching rate (the number of approaches from a juvenile to its adult tutor within 5 cm of distance, per 6 min recording) did not correlate with tutor singing-induced approaching rate (number of juveniles approaching immediately after tutor singing/number of tutor songs) or similarity match to the tutor song (Supplementary Figure 3a, b). Similarly, closer social bonding between a juvenile and its tutor (measured by the accumulative duration between an approaching juvenile and its tutor staying together within 5 cm) did not correlate with singing-induced listening/approaching rate (Supplementary Fig. 3c, d).

To determine the role of the social tutor in the juvenile's listening/approaching behavior, the father tutor was removed in early life (30 dph) and replaced with a speaker playback of the same tutor song ($n = 6$ birds) daily from 30 to 65 dph. During the peak of listening/approaching behavior (48–53 dph), song playback induced significantly fewer approaching responses from juveniles. Juveniles approached the speaker less frequently (i.e., lower approaching rate Fig. 2a), more slowly (i.e., longer time latency, speaker 4.3 ± 0.4 vs. live tutor 2.7 ± 0.6 sec), and in farther distance to the speaker, when compared to the approach toward a live tutor (speaker 11.83 ± 2.07 vs. live tutor 4.35 ± 0.76 cm).

**Listening approaching behavior is selective, attentive, and driven by previously acquired tutor memory**. The time-sensitive juvenile listening/approaching behavior may require previously acquired memory of the tutor song, based on several lines of evidence. (1) When juveniles were tutored by a social tutor (father or foster tutor) from 0 to 65 or 20 to 65 dph, they performed a robust approaching/listening behavior at 44–55 dph when continuously exposed to the same adult tutor (Fig. 1b; $n = 33$ birds). (2) If a father tutor was temporarily removed from 35 to 45 dph ($n = 8$ birds), juvenile males exhibited robust approaching/listening behavior at 46–55 dph when they were re-exposed to the same father tutor at 46 dph (Fig. 2a). (3) If juveniles were not exposed to an adult tutor during the early sensory period (i.e., father tutor removal between 10 and 45 dph), they had significantly reduced tutor-approaching behavior after the father tutor was later introduced at 46 dph (Fig. 2a, Mann–Whitney U test, $P < 0.001$; $n = 8$ birds). The juveniles still exhibited attention-like behavior (such as vocal silence) in response to the tutor's

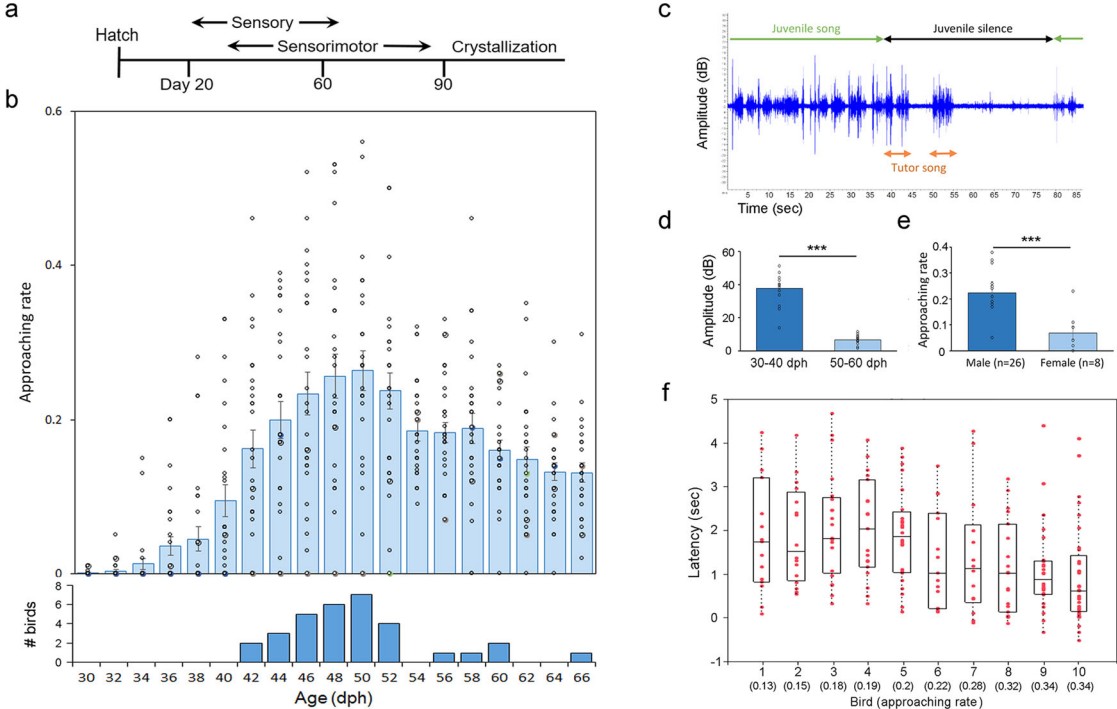

**Fig. 1 Characteristics of listening approaching behavior. a** Sensitive period of vocal learning in zebra finches. **b** Developmental trajectory of attentive listening behavior. Juvenile males ($n = 33$ birds) were housed together with their social tutors during the sensitive period of vocal learning (0–65 or 20–65 dph). The attentive listening and approaching behavior slowly emerged around 38–40 dph, peaked at around 46–52 dph, and gradually declined afterward. The approaching behavior was quantified as approaching rate (number of approaches toward a tutor/number of tutor songs). Lower panel: Age (in dph) of peak approaching behavior (or highest approaching rate). Most male juveniles had their peak approaching behavior around 46–52 dph. Blue bars represent the mean and standard error of the average approaching rate from 33 birds; each black dot depicts the mean approaching rate of each individual bird at a given age. **c** An example shows that, immediately after the onset of tutor singing, juvenile siblings who were singing the plastic song (i.e., emergence of recognizable syllables) became strictly silent. The temporary vocal silence lasted up to 53 s ($23.5 \pm 6.7$ sec, mean ± SEM; $n = 12$ birds at the age of 52–58 dph) before singing resumed. **d** The vocal silence (at least 1 s) occurred only during the late sensitive period (50–60 dph). During the early sensitive period (30–40 dph), juveniles rarely silenced vocalizations (i.e., subsong) after the tutor singing ($n = 12$ birds, Mann–Whitney $U$ test, $P < 0.001$). **e** Sex differences in the listening approaching behavior. Compared to the males, juvenile females had significantly fewer tutor-approaching movements following tutor singing ($n = 8$ females and 26 males, approaching rate was averaged from 45 to 60 dph; Mann–Whitney $U$ test, $P < 0.001$). **f** Individual variation in the time latency of approaching movement. Ten juvenile males were collected from their peak days of attentive listening approaching movement (48–53 dph). Individual birds who had earlier onset (shorter time latency after tutor sings) of approaching movement also had a higher approaching rate (#approaching movement/# tutor song, $R^2 = 0.72$). The approaching rate of each bird is shown on the bottom of the X-axis (in parentheses). For juveniles with a higher approaching rate (Birds #7–10), the initiation of approaching movement began even milliseconds before the onset of the tutor song. Each red dot represents one approaching movement in response to the tutor singing.

singing, but they had much less approaching behavior. The results were consistent with a previous study[13]. (4) If the father tutor and a later introduced adult male stranger were both presented in the same cage with the juveniles after 45 dph, juveniles unanimously approached the father tutor's singing (approaching rate toward father vs. stranger = 0.19 vs. 0.002; $P < 0.001$, $n = 8$ birds). (5) To rule out the possible social influence of father tutors or strangers on the juvenile behaviors described in 4, we played back the tutor song or stranger adult song to the juveniles who were previously exposed to the same tutor song before 45 dph. After tutor removal at 46 dph, tutor song playback at 50–55 dph induced a significantly higher approaching rate ($0.057 \pm 0.013$; $n = 8$ birds) than playback of a stranger song ($0.011 \pm 0.006$; Mann–Whitney $U$ test, $P < 0.01$), as most juveniles rarely approached the speaker of stranger song playback.

**The attentive listening approaching is associated with vocal imitation capability.** The juvenile males (with continuous tutor exposure from 0 to 65 or 20 to 65 dph, shown in Fig. 1b) who had a higher listening/approaching rate during the "plastic song" stage (46–55 dph) also developed better tutor imitation in their

crystallized song (Fig. 2b, $R^2 = 0.56$, $n = 23$ birds). Additionally, juveniles who were continuously tutored, or re-exposed to the same tutor, had a higher approaching rate (Fig. 2a) and a significantly better song imitation compared to the juveniles who were tutored after 45 dph (similarity score: $65.2 \pm 3.3$ vs. $51.7 \pm 4.5$), or tutor playback ($48.4 \pm 3.1$) (Fig. 2c, d). These results were consistent with a previous study[8], where juvenile zebra finches were housed together with their father tutors until 35, 50, or 65 dph. The juveniles who were tutored until 65 dph learned best, and copied most song syllables[8,19,20].

**Attentive listening behavior is dopamine dependent.** Attention-associated motor skill learning is regulated by dopamine[21–23]. To investigate the role of dopamine in juvenile listening behavior, a dopamine agonist (nomifensine, 2.5 mg/kg) was administered subcutaneously in the juvenile males who had an average level of approaching rate (20.6%; range: 12–25%). Post-nomifensine-injected birds showed significantly increased attentive approaching movement ($n = 9$ birds in the nomifensine group vs. 8 birds in the saline-injected group, Mann–Whitney $U$ test, $P < 0.01$; Fig. 3a). The post-nomifensine-injected juveniles approached their tutor more

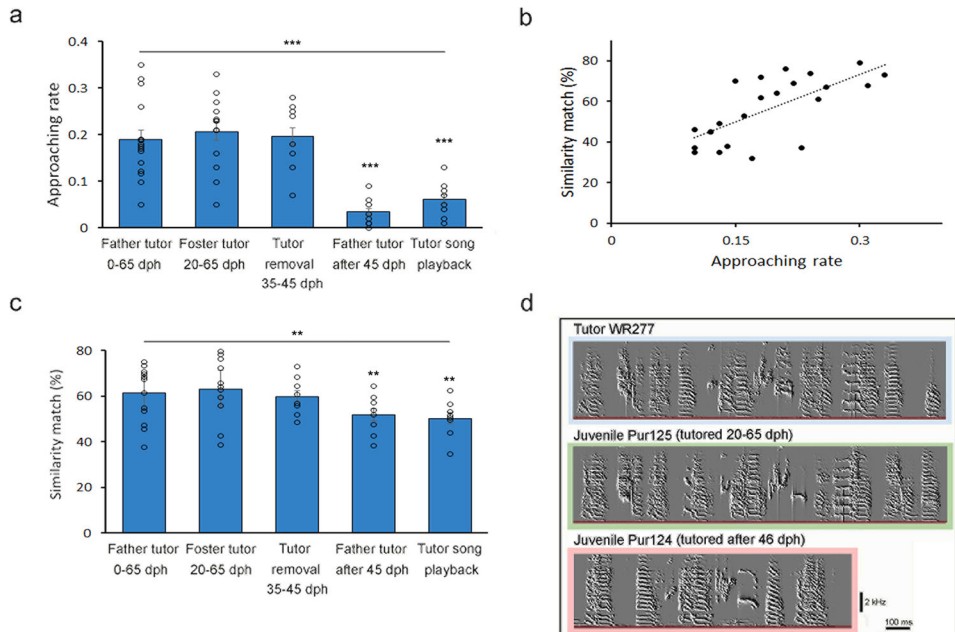

**Fig. 2 Development of juvenile's listening/approaching behavior requires previously acquired tutor memory, and individual variation in attention behavior was associated with the precision of song imitation. a** Juveniles who were socially tutored from 0 to 65 dph or 20 to 65 dph, or experienced tutor removal from 35 to 45 dph and then were re-exposed to the same tutor at 46 dph, had significantly more listening/approaching behavior toward their tutors during 46–55 dph ($n = 23$ birds; one-way ANOVA, $F_{4,23} = 5.71$, ***$P < 0.001$; Tukey post hoc test, tutored 0–65 dph vs. tutored after 45 dph or tutor song playback, $P < 0.001$) than juveniles who were not exposed to adult tutors until after 46 dph ($n = 8$ birds) or juveniles who received the speaker playback of the same tutor song ($n = 8$ birds). Blue bars represent the mean approaching rate with standard error. Each dot represents an individual bird's approaching rate. **b** Juvenile males ($n = 23$, each black dot represents one individual bird) who had a higher approaching rate also developed a better match of tutor song imitation ($R^2 = 0.56$). The approaching rate depicts the mean number of approaches/number of tutor songs. **c** Juveniles who were socially tutored from 0 to 65 dph, 20 to 65 dph, or experienced tutor removal from 35 to 45 dph, had better imitation of tutor song, compared to juveniles who were not exposed to adult tutors until after 46 dph ($n = 8$ birds), or juveniles who received playback of the same tutor song ($n = 8$ birds; one-way ANOVA, $F = 3.83$, $P < 0.01$; Tukey post hoc test of the 20–65 tutored group vs. song playback group, vs. tutored after 45 dph, **$P < 0.05$). **d** A juvenile (Pur125) who was continuously exposed to its father tutor (WR277) from 0 to 65 dph had better imitation of the tutor song than a juvenile (Pur124) who was exposed to its father tutor after 46 dph.

frequently and had an earlier onset of tutor-approaching movement (Fig. 3b, Supplementary Fig. 4, Mann–Whitney $U$ test, $P < 0.001$). Remarkably, some post-nomifensine-injected birds were highly motivated and initiated tutor-approaching movement even before the onset of tutor singing (Fig. 3b, Supplementary Fig. 4), and these juveniles were more likely to peck the tutor's beak after the tutor sang (number of beak pecks/number of tutor songs; before vs. after injection = $0.011 \pm 0.003$ vs. $0.057 \pm 0.009$; Mann–Whitney $U$ test, $P < 0.01$).

Administration of the dopamine antagonist SCH23390 (0.05 mg/kg) to juveniles ($n = 9$ birds, average approaching rate = 19.1%; range: 15–23%) significantly reduced approaching behavior, compared to the baseline and the control group with saline injection ($n = 9$ males, Mann–Whitney $U$ test, $P < 0.001$; Fig. 3c). The juveniles significantly reduced their approaching movement with greater latency of initiating approaching movement (Fig. 3d). Upon hearing the tutor song, post-nomifensine-injected males did not exhibit attentive behavior toward the tutor and they did not freeze ongoing behavior or become silent (Supplementary Movie 4; sound amplitude < 5 s after tutor singing, before vs. after injection = $5.1 \pm 0.8$ vs. $33.7 \pm 6.2$ dB).

**Forebrain correlates of juvenile listening approaching behavior.** To explore the neural correlates of the listening approaching behavior, the neural activity-dependent immediate early gene, *Egr1*, was used to identify the forebrain neural activity associated with the juvenile approaching behavior. The attentive approaching behavior was associated with mRNA expression of

*Egr1* in the caudal medial nidopallium (NCM), and frontal nidopallium (NF). The expression level was significantly higher in juveniles who approached toward a previously exposed adult tutor, compared to juveniles exposed to novel adult tutors, or playback of father tutor song (Fig. 4b). Moreover, the juveniles who showed higher attentive approaching behavior also had higher *Egr1* expression in the NF than less attentive siblings (Mann–Whitney $U$ test, $P < 0.01$). The *Egr1* expression level in those regions was independent of the amount of the juvenile's plastic song singing (two-way ANOVA: the amount of singing and approaching behavior; $P < 0.05$). It has been suggested that the NCM encodes the auditory memory of the tutor song[24] and the NF involves both auditory processing and somatosensory function[25].

## Discussion

Our results provide observational and experimental evidence that juvenile songbirds develop a phase-sensitive auditory learning program, as juvenile zebra finches actively and selectively engage in listening/approaching behavior toward a previously exposed tutor song during the sensorimotor integration, or plastic song stage. This specialized listening and approaching behavior is likely driven by previously acquired tutor song memory. The execution of attentive approaching behavior is possibly triggered by both visual and auditory stimuli of the singing adult tutor, as the more motivated juveniles paid higher attention to the tutor behavior and initiated approaching behavior even before the onset of tutor singing. Vocal learning in a more natural condition may thus

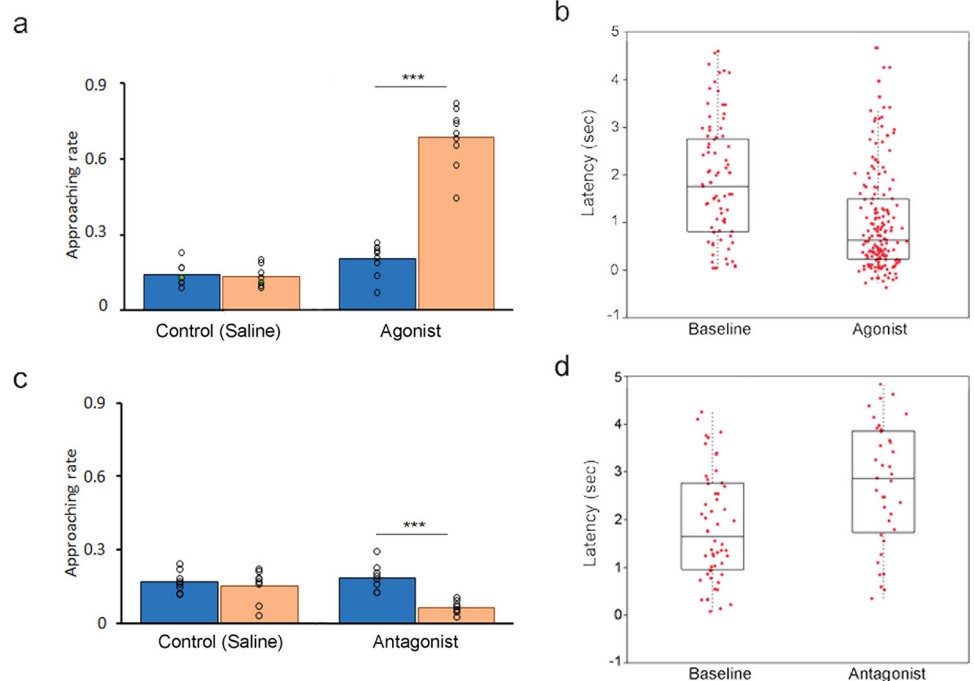

**Fig. 3 The administration of dopamine agonist and antagonist had a significant effect on juvenile listening approaching behavior. a** Subcutaneous administration of a dopamine agonist (nomifensine, 2.5 mg/kg) at 50–55 dph. Post-nomifensine-injected birds (orange bars with standard errors) showed a significantly increased approaching rate (# approaching movement/# tutor song) after the tutor sang, compared to the pre-injection period (blue bars with standard errors) or saline-injected controls ($n = 9$ birds in nomifensine vs. 8 birds in the saline-injected group, Mann–Whitney $U$ test, $P < 0.001$). **b** Post-nomifensine-injected birds approached their tutor more frequently and had an earlier onset (shorter latency) of approaching movement ($P < 0.001$). Boxplots show the distribution of time latency to initiate approaching movement from nine post-nomifensine-injected birds. In each boxplot, the lines from the top to bottom were defined as the third quartile, median, and first quartile. Each red dot is one single recording of the approaching movement. Each individual's post-nomifensine-injection response (approaching latency) can be seen in Supplementary Fig. 4. **c** Administration of a dopamine antagonist, SCH23390 (0.05 mg/kg; $n = 9$ birds), showed significantly inhibited attention and approaching behavior, compared to the saline injection ($n = 8$ males, Mann–Whitney $U$ test, $U = 2.17$, $P < 0.001$). **d** Post-SCH23390-injected juveniles showed significantly reduced approaching movement with greater latency to initiate the approaching movement.

involve complex and multiphasic social interactions between the tutor and the tutee.

Similar but weaker attentive listening behaviors have been reported in a number of studies[15,26,27]. However, none of these studies described a highly motivated tutor-approaching behavior, possibly because these studies were conducted under various manipulative conditions. For example, attentive behavior was also observed by Chen et al.[15], where the juveniles who were not tutored until after 40 dph exhibited attentive but no approaching behaviors, which is consistent with our tutor-deprivation results (Fig. 2a) and supports our hypothesis that listening approaching behavior requires earlier-acquired tutor song memory. Additionally, in an operant conditioning experiment[27], juvenile finches substantially increased lever-pressing for song playback around 50 dph, similar to our observations of the peak approaching rate around 46–55 dph. Interestingly, as soon as the juveniles pressed the lever for song playback, they were likely to fly over and peck the speaker. Similarly, in our study, more motivated juveniles are likely to fast approach the tutor and peck the tutor's beak in response to the tutor singing.

What might be the adaptive function of attentive approaching toward a specific tutor during the sensorimotor learning or plastic song stage? Zebra finches are colonial species. In nature, free-living juveniles have access to many adult males during the sensitive period of song learning, and yet young finches usually commit to only one tutor, typically their father, from whom they precisely imitate the song[28]. Precise vocal imitation to a specific tutor may be crucial for effective social communication among close neighbors or colony members to increase reproductive success in nature[13,29]. The attentive listening/approaching behavior during the plastic song stage may provide an important function to facilitate a juvenile's instant access and selectively attend to a re-exposed or repeatedly exposed adult tutor and fine-tune its previously memorized "template" during sensorimotor integration.

The active and attentive approaching/listening behavior by juvenile zebra finches may serve one and/or two possible functions in the developing vocal learning circuits: (1) attentive approaching may activate or strengthen the previously acquired tutor memory and allow for more precise vocal imitation, and we speculate the listening/approaching behavior may strengthen forebrain auditory memory circuits[24,30]; (2) attentive approaching may selectively facilitate the conversion of a previously acquired auditory memory to vocal motor output. The song circuit, including cortical-basal ganglia dopaminergic pathways, for sensorimotor integration may be strengthened[22,31,32].

We propose vocal learning under a natural condition may go through a two-step auditory process. The juvenile may first passively acquire or imprint the tutor song memory during the early sensitive period. Then, during the later phase of sensorimotor integration, the previously acquired model memory is used to guide them to selectively attend to and anticipate the song of a specific tutor and fine-tune vocal imitation by developing active listening approaching behavior. We suggest that the early imprinted model memory or template through social interaction may thus serve an adaptive function for the later demand of

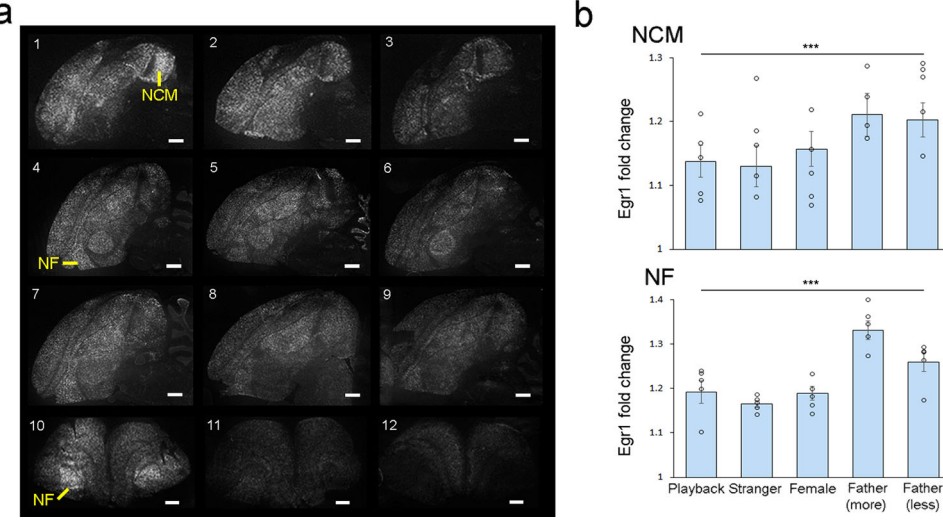

**Fig. 4 The listening approaching behavior of juvenile zebra finches was associated with specific forebrain regions.** Neural activity-dependent marker, *Egr1*, was used to identify the associated forebrain region (**a1–a9**). Sagittal view shows higher *Egr1* mRNA expression in the caudal medial nidopallium (NCM) of juveniles in response to the father tutor's song (**a1**), compared to the playback of the tutor song (**a2**), and singing of a stranger adult (**a3**). Similarly, higher *Egr1* expression was identified in the frontal nidopallium (NF) of juveniles who showed a higher approaching rate toward father tutor singing, regardless of the amount of juvenile singing (**a4, a7**), compared to the juvenile siblings who showed a lower approaching rate toward father tutor (**a5**), approaching rate toward the tutor song playback (**a6**), approaching rate toward singing of a stranger adult (**a8**), or juvenile female's response toward father's singing (**a9**). (**a10–a12**) Coronal view shows a higher *Egr1* expression in the NF of juveniles who exhibited higher levels of approaching behavior toward the father tutor (**a10**), than in non-approaching juvenile females (**a11**) and in juveniles singing in social isolation (**a12**). Scale bar = 800 μm in **a1–a9**; and 350 μm in **a10–a12**. **b** *Egr1* mRNA expression had significant differences in the NCM and NF among five experimental groups (playback of tutor song, $n = 5$; exposure of a newly introduced stranger adult, $n = 5$; juvenile female's approaching toward father tutor, $n = 5$; juvenile males with high and low approaching rate toward father tutor, $n = 7$ each; one-way ANOVA, $F_{5,26} = 4.67$; Tukey's post hoc test, father tutored (0–65 dph) males vs. father tutored females; father tutored (0–65 dph) vs. playback of father tutor song; vs. stranger adults. ***$P < 0.01$). Blue bars represent the mean *Egr1* fold change with standard error. Each dot depicts the *Egr1* expression level (fold change) of each individual bird.

selective attention, decision making to enhance more precise sensorimotor skill learning.

The juvenile auditory learning strategy that we described here may be common among other seasonal songbirds as well. Growing evidence from both field and laboratory studies suggests that juvenile songbirds usually first acquire tutor song memory during the early sensory learning phase, but then are re-exposed to the same or a similar tutor song during the later sensorimotor integration phase, through an instructive or selective attrition strategy[10–12,15,29,33]. Re-exposure or repeated song exposure from a selected adult tutor may ensure the final commitment of song acquisition during territory settlement[12,28]. Precise tutor imitation from a selected tutor seems critical to enhance social communication in a dynamic social environment[13].

## Methods
### Animals and behavioral studies

*Animals.* A total of 196 male and female zebra finches (*Taeniopygia guttata*) from Colgate University's animal facility were used in this study (Supplementary Table 1). Some adult tutors were used for more than one clutch. Each clutch consisted of one adult male tutor and one or two juvenile males (see below for details) housed in a medium-sized cage (35 × 52 x 61 cm). Birds had access to seed, water, vegetable, egg supplement, and grit ad libitum and were on a 12:12 light cycle from 0800 to 2000. This research has been approved by the IACUC committee at Colgate University.

*Behavior monitoring and recording.* The social interaction and juvenile listening behavior were video- and audio recorded using wi-fi network cameras (D-link DCS 942 L, CA, USA) positioned above each bird cage. The wire tops of the bird cages were removed and replaced with transparent plexiglass to allow for maximal visibility. Video recordings were collected from 0800 to 1000 when the juvenile birds were judged to be most active in singing and listening movements. Videos were recorded every other day during juvenile aged 30–65 dph. This age is consistent with the approximate onset of sensorimotor learning[1]. Juveniles and tutors in the same cage were marked using different colors of acrylic paint that allowed for visual identification of each individual and tracking the movement from video

recordings using a tracking software (EthoVision XT 14, Noldus). All of the experimental data were analyzed by manual scoring aided by the tracking software (see below for details). Each subject's head crown was painted a distinct color to easily differentiate the tutor and each juvenile during viewing or tracking (Supplemental movie 1). Four student scorers independently and manually rated the approaching behavior based on the color paints of each individual bird.

*Quantification of singing-induced listening/approaching behavior.* From each video recording, four student scorers manually scored approaching/listening behaviors by viewing the video recording files. The student scorers had no previous knowledge of zebra finches, their age, or gender differences in morphology. The scorers had an almost unanimous agreement (>93%) in deciding what was approaching behavior, attentive but no approaching, or other social interactions. As mentioned in Results, we define a juvenile's listening approaching behavior as immediate orientation and flying in close proximity (<5 cm) toward its tutor upon hearing the tutor song (<5 s after the onset of tutor singing). The approaching behavior is immediately followed by temporary freezing of ongoing behavior (such as flying, feeding, and preening) and strict silence for at least 1 s. To better quantify the distance (proximity of <5 cm) between a tutor and its approaching tutee, a tracking software (EthoVision XT 14, Noldus) was used to aid manual scoring. The software allowed the scorers to track and measure the approximate distance between an approaching juvenile and its tutor. Additionally, the tracking software allowed scorers to slow down the video frames and manually examine the approximate timing (onset and duration) of approaching behavior, and the duration of vocal silence or freezing behavior.

*Quantification of juvenile approaching movement under all social contexts.* To measure the overall approaching behavior of a juvenile to its tutor under all social contexts (tutor singing or not), we used a social proximity function from a movement tracking software (EthoVision XT 14) to automatically track the occurrence of approaching between an approaching actor (juveniles) and a receiver (tutors). We set up the proximity threshold between the approaching actor and the receiver as within a distance of 5 cm. The software can automatically quantify (1) the overall approach rate (number of juvenile approaches per 6 min recording) and (2) the mean cumulative duration of approaching (percentage of time period when two birds stay within 5 cm of distance, per 6 min recording) during the sensitive period of song learning (30–65 dph). We then compared the correlation between the overall approaching rate and the singing-induced approaching rate. Totally 17 males who had complete video recordings were used in this analysis. In our

preliminary test, the data analyzed by the software were later confirmed by our manual scoring. This automatic tracking software, however, does not detect sounds and is not able to distinguish singing-initiated approaching behavior from other non-singing-related approaching behavior.

*Quantification of singing-initiated beak pecking.* We define beak pecking as when a juvenile pecks his tutor's beak immediately after the end of tutor song (< 5 s). To quantify the frequency of singing-related beak pecking, the scorers viewed video recordings (see above) and manually scored the number of beak pecks following each tutor song. We then used Pearson's correlation coefficient to test the correlation between the frequency of singing-associated beak pecks (number of beak pecks/number of tutor songs) and juvenile approaching rate (number of approaches/number of tutor songs).

*Tutor removal experiment*

### Father tutor removal between 35 and 45 dph.
Juveniles (n = 8) from four clutches were used for this experiment. In each clutch, the father tutor was removed when the juveniles were aged 35–45 dph, and the juveniles were housed alone where they could not see, hear, or socially interact with any adult males. At 46 dph, the father tutor was returned to the juvenile cage. Juveniles' social interactions with the father tutor were recorded until 65 dph. Each juvenile was then housed individually in a sound-proof chamber and their song was recorded again at the age of 100 dph to measure the song similarity matching score between tutor and tutee (see below).

### Father tutor removal from 10 to 45 dph.
Juveniles (n = 8) from five clutches were used for this experiment. In each clutch, the father tutor was removed when the juveniles were aged 10–45 dph, and the juveniles were raised by the mother alone. The juveniles and their mother were housed in a room where they could not see, hear, or socially interact with any adult males. At 46 dph, the father tutor was returned to the family cage. Juveniles' social interactions with the father tutor were recorded until 65 dph. Each juvenile was then housed individually in a sound-proof chamber and their song was recorded again at the age of 100 dph.

*Playback experiment*

### The effect of playback or social tutoring on listening/ approaching behavior.
Six juvenile males from three clutches were used for the song playback experiment. The juveniles and their tutors were kept together in a medium-sized cage until 30 dph. The father tutor was then removed and the juvenile was housed in a cage with acoustic and visual isolation from other birds. The listening behavior was video-recorded every other day from 0800 to 1000 as described earlier, from 31 to 65 dph. A speaker was placed next to a perch. The songs from the same tutor were played back for 30 min daily at a similar singing rate to that of live tutors in the morning at 0900. The tutor song was sampled from recordings of ~ 30 different song renditions and synthesized in a 30-min section, and each 30-min playback section had 180–190 songs (Raven 4.1, Cornell University, NY).

### Playback of tutor song vs. stranger song.
Eight juvenile males from four clutches were used for the playback experiment. The juveniles and their tutors were kept together in a medium-sized cage until 45 dph (see above for details). The listening/approaching behavior was video-recorded every other day from 0800 to 1000 as described earlier. For each clutch, the adult tutors were removed at 46 dph. A speaker was then placed next to a perch of the cage. The songs from the same tutor or a stranger were assigned randomly each day and played back for 30 min in the morning at 0900, during the peak of approaching behavior at around 50–55 dph. The playback song was prepared at a similar singing rate and amplitude to that of live tutors. To avoid pseudo-replication, a stranger song was recorded from eight adult males, and one of the stranger adult songs was played back to each juvenile. During each playback section, the approaching movement of the juveniles was recorded by a camera (see above for details), and the video recordings were later analyzed manually.

*Song recording and similarity scoring.* The crystallized song of experimental and control juveniles was recorded at ~ 100 dph. Individual birds were placed in isolated recording chambers and their songs were recorded and analyzed using Song Analysis Pro 2011 software (SAP)[34]. Song similarity was measured using SAP. For each tutor–tutee pair, ten tutee motifs recorded at 100 dph were sampled and compared to ten tutor song motifs using asymmetrical time courses. The similarity scores for each group of tutor–tutee song comparison were averaged for statistical analysis of song similarity between the pair.

*Statistical analysis.* One-way ANOVA was used to compare the statistical differences among experiment groups (tutor removal, father tutor, and playback groups), we used the Tukey post hoc test to further compare the significant difference between groups. Mann–Whitney U test was used to test the significant differences between two experimental groups (i.e., sex differences in approaching rate, age

differences in silence responses toward tutor song in Fig. 1d, e). Pearson's correlation was used to compare the correlation between two variables (i.e., approach rate vs. similarity matching in Fig. 2b; approaching rate vs. number of beak pecking/ tutor song in Supplementary Fig. 1; and approaching rate vs. overall approaching in Supplementary Fig. 3).

### Administration of dopamine agonist and antagonist.
Juvenile males (n = 35 birds) from 15 clutches were raised by biological parents until ~ 30 dph. Fathers and juvenile males were subsequently moved to medium-sized cages (as described above) for video–audio recording and monitoring of listening approaching behavior. Each clutch had one father and two or three juvenile sons for the study.

*Behavior monitoring and scoring.* In order to measure this listening/approaching behavior, network video cameras were used for recording between 30 and 65 dph. Juveniles' behavioral responses to the tutor's song were manually scored by three scorers. We used a tracking software (see above) to manually calculate the distance of movement, the duration, and onset timing of the approaching behavior. More specifically, each juvenile was marked as approaching or non-approaching, based on the definition of listening/ approaching behavior described in the main text. After all songs and behaviors were recorded for a specified time span of 2 h, the number of times each bird approached, number of times each bird attended, and number of times the father sang were all tallied. These quantitative analyses allowed us to measure approaching rate and latency, and test our hypothesis that attentive approaching behavior is associated with dopamine. We recorded pre-intervention for 3 days prior to drug administration.

*Pharmacology Protocol.* The effects of a dopamine agonist and antagonist on listening approaching behavior were tested using subcutaneous injections into the inguinal region of the male juveniles. The dopamine agonist and the dopamine antagonist were injected at the peak period of attentive listening (~ 46–55 dph).

Nomifensine (MedChemExpress, HY-B1110A), the selected dopamine agonist, increases extracellular levels of dopamine and norepinephrine by blocking functioning of the NE/DA-T[35] and has been used as a DA reuptake inhibitor in birds[36]. Nomifensine has been shown to greatly increase extracellular levels of DA in male zebra finches[37]. SCH23390 (Sigma, D054), the chosen dopamine antagonist, selectively blocks the D1 receptor subtype has been used in various animal models[19].

The injection protocol for our experiment followed a 3-day time course. For three consecutive days, all subjects were subcutaneously injected in the inguinal region at 1000 and behavioral data were recorded for 1 h immediately before injection (0900–1000) and 2 h post-nomifensine-injection (1030–1230). For each experiment, one or two juvenile siblings per clutch were assigned to the drug condition (n = 9) and the remaining one was assigned to the saline-control condition (n = 8). All subjects within the drug conditions were drug-naive for each experiment. For the dopamine antagonist experiment, each day the drug condition received 0.2 mL of 0.05 mg/kg SCH23390 and the control condition received 0.2 mL of saline. Based on preliminary observation, a 0.05 mg/kg dose of SCH23390 typically lasts 2 h. For the dopamine agonist experiment, each day the drug condition subjects received 0.1 mL of 2.5 mg/kg nomifensine and the control subject received 0.1 mL of saline. Preliminary observation shows that the selected dose of nomifensine had effects lasting up to an hour.

*Analysis of behavioral data.* Both experiments employed the aforementioned behavior scoring protocol. Over the drug manipulation period, subjects' behaviors were recorded to investigate the effects of the selected drugs on their attentive-listening behavior and subsequent song learning. Every instance of tutor singing during those recordings was separated into small video clips for future behavior scoring by third-party raters. For both experiments, juveniles' behavioral responses to the tutor's bout of song were rated by three students blind to the subject's condition. For the purpose of our study, we chose to focus on the category of approaching and denote categories of attending and neither as not approaching. Approaching suggests greater motivation than simply attending and may even be the stronger form of attentive listening. An intraclass correlation coefficient and its 95% confidence intervals were calculated using SPSS statistical package version 24 (SPSS Inc, Chicago, IL) based on a mean-rating, absolute-agreement, and two-way mixed-effects model. Mann–Whitney U test was used to compare the significant difference between baseline and post-nomifensine-injection groups.

### Neural correlates of listening approaching behavior
*Behavioral monitoring.* Seven clutches of zebra finches (31 birds in total) were used in this study. In each clutch, there were two male siblings (or one male and one female sibling) and they were housed together with the father tutor in a medium-sized cage. The individual's attentive approaching behavior was video-recorded and visually monitored for 1 h after lights were on (0800), from 40 to 60 dph. We made sure that the adult tutor produced at least 20 songs from 0800 to 0900 and the number of father songs was recorded. The siblings (with the higher or lower approach rate) were then sacrificed at about 0910. (a) Approaching the father tutor. Fourteen juvenile males were used in this study; during the peak approaching period (46–55 dph), we recorded the juvenile approaching behavior from 0800 to

0900 and then sacrificed the birds at around 0910. (b) Approaching a stranger. During the peak of individual approaching, a stranger adult male was introduced as a novel tutor to the juveniles, and their father tutor was removed. Tutor's singing and juveniles' approach behavior ($n = 5$ birds) were recorded and juveniles were sacrificed following the procedure described in (a); (c) Approaching song playback speakers. Each of five juvenile males was housed singly during the peak of the approaching behavior (45–55 dph) and the father tutor was removed. The father tutor song (~ 20 different song renditions recording from each tutor, see the above methods for playback experiment) was played back to its juvenile for 1 h (0800–0900). Juveniles were then sacrificed at ~ 0910. The brain tissues were extracted and stored in a −80° C freezer for in situ hybridization.

*In situ hybridization*. In situ hybridization followed the protocols previously described[38,39]. In brief, frozen brain sections (20 μm) were hybridized with digoxigenin-labeled and confirmed by 35S-labeled anti-sense riboprobes with modifications in processing post hybridization of digoxigenin-labeled slides. After washing with 2xSSC (saline sodium citrate), the slides were placed in TN buffer in 10% non-fat milk, then applied 1:200 dilution of anti-Digoxigenin-AP, -fab fragment (Sigma) overnight. The slides were then washed with TN and TNM buffer, and incubated in NBT/BCIP solution (Sigma) overnight. Gene expression level in the specialized forebrain song nuclei region was then quantified by using the brain image on the exposed slides, or the films were placed and scanned at 7000 dpi (Epson, Perfection V700, Long Beach, CA). Images were then exported to Adobe Photoshop CS3 (Adobe, San Jose, CA) and converted to 8-bit grayscale. The forebrain regions and surrounding areas were outlined, and the average pixel density was calculated using the Photoshop histogram function. anti-Digoxigenin-AP, Fab fragment

For image analysis of NCM, three sections (per hemisphere) at the medial parasagittal position (between 180 and 330 μm from the midline) were collected. The ventricle was used as dorsal, caudal, and ventral boundaries. For measurement of NF, three sections from the parasagittal position (1200 μm from the midline) and four coronal sections (970–1350 μm from the rostral end) were collected from each bird. LaM (Lamina mesopallialis) was used as dorsal, medial, and lateral boundaries, and Bas (Nucleus basorostralis pallii) as ventrocaudal boundary. All data were first normalized to the mean level of *Egr1* expression in the silent condition ($n = 3$ birds) at the same age. Each slide was covered by a tape and the gene expression levels were measured by two students blind to the slides. To quantify and compare the *Egr1* expression level among different birds and treatment groups, we normalized the number of song bouts produced by each tutor from 0800 to 0900 (after the lights were on at 0800, see above). One-way ANOVA and Tukey post hoc tests were tested for statistical differences among groups.

**Statistics and reproducibility**. The research sample included 167 juvenile male and 12 juvenile female zebra finches. The data in each graph were analyzed with Mann–Whitney *U* test (two groups) or one-way ANOVA (multiple groups) with Tukey post hoc test. Pearson correlation coefficient was used to measure the linear correlation between two variables (experimental groups or conditions). All statistical tests were conducted using SPSS 24. Each individual animal or behavior was displayed as an individual point in the boxplot with median, the first and third quartile, or the bar graph with mean and standard error. Detailed statistical methods in each experiment are described in the relevant methods sections and figure legend. Reproducibility can be accomplished by following the protocols or experimental methods mentioned in the relevant method sections.

**Reporting Summary**. Further information on research design is available in the Nature Research Reporting Summary linked to this article.

## Data availability
The source data underlying main and supplementary figures are presented in Supplementary Data 1-2. Other data are available from the corresponding author upon reasonable request.

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

## Acknowledgements
We thank Cindy Baker and the animal caretakers from Colgate University for their wonderful help and bird care. This research is supported by Picker ISI research grant to WL.

## Author contributions
W.L. designed the study. W.L., M.L., G.S., M.I., and F.F. collected the data. W.L. M.L., G.S., M.I., and F.F. analyzed the data and prepared and reviewed the manuscript.

## Competing interests
The authors declare no competing interests.
