## [Transparent Peer Review File · Communications Biology]

Reviewers' comments:

Reviewer #1 (Remarks to the Author):

Liu et al carefully observe the movements of zebra finch clutches as juvenile birds acquire songs from tutors. They discover that juveniles tend to approach tutors right around the time they begin to sing and that this behavior peaks around the period of development that young males are primed to learn the song. The approaches happen within seconds (sometimes less) of when a tutor starts to sing, and the approaches are specific to a previously 'chosen' tutor bird. Females do not exhibit this behavior. Interestingly, approach behavior is correlated with imitation quality. These are exactly the type of careful behavioral experiments that are necessary to advance the field - they open up many future opportunities to identify neural mechanisms of tutor selection and song learning. The authors make initial inroads into mechanism by showing that activating or suppressing dopamine signaling (systemically) can influence approach behavior. They also show that immediate early gene expression in NCM, an auditory area, correlates with approach behavior. This is an important advance for the field and I support publication provided some minor issues are addressed.

(1) Individuals who are peak approaches approach less with DA antagonist. Individuals who are lousy approachers approach more with DA agonist. The authors do not spell out what happens when drugs were given to 'normal' birds at median levels of approach behavior - and it seems odd that they only did the agonist experiments on the low-quality approachers and the antagonist experiments on the high quality ones. Was this a post-hoc decision to specifically pull out the part of the dataset where there was a significant effect or were these the only DA drug experiments performed?

(2) Multiple sections of the paper, including the videos, refer to beak-pecking as common mode of interaction during tutoring. Tutors peck 'inattentive' juveniles and juvenile peck tutors. But this interesting behavior is only barely. In line 56 it says $r=.63$, Person's coefficient) but it is not stated in the methods what exactly was correlated with what. In addition, pecking rates and tutor singing rates are reported to be changed in the dopamine experiments but these data are not quantified.

Reviewer #2 (Remarks to the Author):

Liu et al. base their experiments on the assumption that song learning in male zebra finches involves a two-step process. A sensory period, during which a song memory is formed and a subsequent sensory-motor phase, during which the vocal output is matched with the memory in an iterative process. In the present manuscript they hypothesize that for the second phase an active-listening behavior or the juvenile males is important and depends on prior exposure to song during the sensory phase. They therefore (1) quantify the percent of time that juvenile males approached the adult tutor during the song learning phase from 30 to 68 days after hatching. They found that peak of approaching behavior centered around 50 days of age, coinciding with very active plastic song production of the juveniles. They also report (2) that juveniles exposed to a tutor only after 45 days of age approach the tutor, that is introduced late, much less than juveniles that are co-habiting with the tutor between 0 and 65 days of age. The juveniles exposed for longer time also develop better song copies than the juveniles exposed only after 45 days of age. (3) Using pharmacological manipulations, the dopamine agonist nomifensine and antagonist SCH23390, altered the birds 'active listening behavior'; injection of the agonist more than tripled the approach rate, whereas the antagonist-injected birds approached the tutor by a third less than saline-injected juveniles. (4) The authors also provide data on the immediate early gene *egr-1* expression in two brain regions, NCM and frontal nidopallium, which differed among groups. The authors conclude from this that the listening-approach is selective and driven by previously acquired tutor memory. The individual findings are interesting. There is increasing evidence that social interactions between

juvenile songbirds and their father/tutor and their mothers and siblings impacts song production learning in males and song preferences and mate choices in females. However, the authors did not frame their study in this context but postulate that 'active listening', e.g. flying towards the singer and suspending their own singing and other activities around 46-52 days post hatch results in better song learning, but only if a song template has already been formed earlier, before the sensory-motor phase of song learning. I find both premises hard to follow. Firstly, Roper and Zann showed in 2006 that in zebra finches there does not seem to be a sensory 'listening only' phase preceding the 'babbling' sensory-motor phase. Both phases fully overlap, listening and rehearsing. Second, the data of approaching the singer and beak pecking are hard to evaluate without knowing how often a juvenile approaches the tutor/pecks his beak outside of the song context. Birds fly/hop around constantly and do a lot of beak pecking. In order to know whether juveniles approached tutors more as a result of the tutor singing I would think that it is necessary to know how often juveniles approached tutors during the times that tutors were not singing. Clearly, approaching and beak pecking could serve multiple behavioral functions, so if there was not difference between tutor-singing and tutor-silent it would not devalue the data but if there was a difference it would certainly help to convince me that the approaching and beak pecking were related to song learning. If there were no difference, it could still be that birds approach the tutor for different reasons, when he sings to listen and when he does not sing for other reasons or pure chance. But it is hard to evaluate the reasons why a juvenile flies towards the tutor without more information on chance/different reasons. The fact that the cages (in the videos) seem to be somewhat small and opportunities for sitting are limited does not aid in this evaluation. A larger cage with more places to sit would make it easier to evaluate whether an approach towards the tutor was deliberate or chance. The fact that birds suspend vocalizations for around a minute after they join the tutor is consistent with the notion that they are listening, but one would also like to have more information on whether and when juveniles suspend vocalizations for a minute outside of the 'listening context'. If they never do this except when 'listening' to the tutor singing, this would be more convincing.

In addition, many of the experiments have other confounds. For instance, introduction of an unfamiliar adult male does not only change the behavior of the juvenile because he might not have been imprinted on him in an earlier phase but the unfamiliar male might also behave more aggressive towards the unfamiliar juvenile. Likewise, exposing a juvenile to a tutor only at day 45 not only prevents the early imprinting phase but also changes the total amount of song exposure. The tutor who has not been exposed to his son for the last 35 days might also have a substantially different behavior towards his son that had they cohabited the entire time. Finally, immediate early gene expression, at least in NCM, might be affected by the total amount of song heard in the different groups, which was not quantified as far as I could gather.

In summary, the different findings are each interesting and some experiments involved a large number of birds, and groups often consisted of sibling cohorts to control for genetic effects, which is very nice. However, the premise of paper (two phases of song learning) is not substantiated by previous data and moreover the story that the authors construct around the individual behaviors does not hang sufficiently well together, given that alternate explanations are possible for each experiment. The manuscript would benefit from major revisions, see specific points below.

Line 24 „Vocal learning in songbirds is generally considered a two-step learning process (Fig. 1a).“ Vocal learning in songbirds follows very different trajectories depending on the species (see Supplemental Info Box in the cited article Bolhuis et al 2010). The time line of the x-axis in Fig 1a suggests zebra finches, but in zebra finches sensory and sensory-motor phases overlap completely (Roper and Zann, 2006). Line 29: ‚restore‘ should be ‚store‘? Line 32: „Moreover, the growing number of field studies suggest that, even if model memory is first acquired (*my italics*) during the sensory learning phase, later re-exposure of the model during the sensorimotor learning phase is critical to ensure selective attention on final song commitment and precise model imitation during territory establishment (8-10).“ ‚Acquired‘ sounds very much like ‚learned‘ but Reference 9 (Liu and Nottebohm 2007) shows clearly that the ‚precursor‘ songs of chipping sparrows are not acquired during a sensory learning phase with reference to an external model. Instead, it is the later exposure to a model/neighbor that causes one of the precursor songs to be modified to the final version. Rephrase

to be more clear.

Line 69; "peaks" should be past tense "peaked"

Line 79: legend to figure 1f) missing info what the red line depicts

Line 90: "The juveniles who had a higher approaching rate also approached their tutor faster (i.e., shorter time latency between the onset of tutor song and tutor- approaching, Fig. 1f)." This sentence implies that there is a correlation between approaching rate and latency, but Fig 1f does not show this.

Line 96 "sit up gesture" (Supplementary Movie# 4)." Please indicate in the movie when the 'sit-up gesture' starts and ends. Alternatively, provide a series of still photos. I am not sure when the sit-up is happening.

Line 124 Supplementary Figure 2. "There was no correlation between an individual's approaching rate and its proximity to the tutor" How was proximity quantified? What does the % on the y axis refer to? Percent time of how many video minutes quantified? And what was the distance definition to qualify as 'proximity'?

Line 146 "males performed robust approaching and listening behavior at 45-55 dph when they were re-

exposed or repeatedly exposed to the same father tutor (Fig. 1b)" Fig 1b does not show data about re-exposure

Line 159 "The results suggested that highly motivated approaching behavior is driven by re-exposed or repeated exposed tutor song memory." I don't think that can be concluded, since there are many other reasons in addition to number of approaches that could cause juveniles that had exposure to their father only after 45 have poor song copying success (less time with the tutor, different social relationship with the tutor, sensitive period no longer optimal, siblings already copying each other during the time without tutor etc)

Line 177 Figure 2 c and d: Given that the 0-65 day group had 20 days more exposure to tutor song than the 0-45 day group it is hard to know whether the lack of interaction with the father or the lack of exposure caused the lower copy success. The 0-45 day group could hear the tutor from 2m away but not see him. One could argue that this acoustic distance and visual isolation might cause the juveniles to categorize the song as no longer socially relevant. This experiment would need to be better controlled in terms of potential variables affecting tutor imitation.

Line 240 "Each red dot is one single recording of the approaching movement." This means that some birds might have contributed many more data than others to plots b and d. I suppose this is for illustrative purposes only and the statistics were done on the averaged data of n=6 birds? If the authors want to keep the graph in panel b and d as is, then I suggest to mark the individual birds with different symbols or colors so that the identity of the birds can at least be eyeballed visually.

Line 251 "The expression level was significantly higher in juveniles who approach" approach should be past tense

Line 262 Figure 4: (b) top and bottom panel not specified which is NCM and which is NF.

Line 328 animals. A table with the animals used in the different experiments would be useful. As it is it is hard to have to go back and forth to find this information, partly in the legends, partly in the text and partly in the methods.

Line 392: is there a reason why the authors used events recording rather than duration ("the number of times each bird approached, number of times each bird attended, and number of times the father sang")

Line 398: what age was the agonist injected? Seems like at a different age than the antagonist. Please state. Possible confound. Injecting unresponsive same age birds with agonist and responsive birds with antagonists would be better, and presenting data as 'before-after' by bird rather than group.

Line 407 ff: it would be nice to see a time line of the assayed behavior for the 1 hour before injection and then three hours post-injection, in addition to the data provided in Fig 3, to appreciate the within animal effect of the drug taking effect and the effect disappearing again.

Line 455 in situ hybridization paragraph: missing a lot of information. As previously described has no reference, borders of NCM, NF and MFV not explained, how many sections were quantified and how, normalization of one group against averages of two other groups unclear how and why. MFV – the abbreviation is not explained and the results for MFV do not appear in the manuscript.

Signed Constance Scharff

Reviewer #3 (Remarks to the Author):

In this study, Liu et al examine the sensory phase of vocal learning in zebra finches, and show that it is composed of two distinct stages: a stage of passive listening and formation of a memory of the learning target, and a stage of active listening and behavioral engagement of a juvenile with an adult "tutor". The authors demonstrate that the latter stage involves approaching and listening behaviors that are directed specifically toward the tutor whose song was heard during the first (passive) stage of learning; that it facilitates vocal imitation; and that it is modulated by dopamine and accompanied by activation of a high auditory forebrain region thought to store auditory memories. The authors hypothesize that attentive and active listening to a previously memorized song serves to fine tune the stored memory of a complex song and thus facilitate precise imitation. The findings are novel and extremely interesting, beautifully showing how juveniles kept with their fathers in semi-natural conditions immediately cease any on-going activities (including their own singing) and approach their father as he starts singing, and sometimes even right before he starts singing (indicating that juveniles are highly attentive to any behaviors that may indicate an intention to sing in the father). The results are robust and convincing. The experimental and analysis methods are adequate (though need some elaboration, see below), and overall the writing and figure presentation are clear and to the point. The study can therefore be a valuable contribution to Communications Biology. I have two main comments that should be easy to address:

1. The method for quantifying the approaching/listening behavior should be described in more detail, both in the methods section and in the results, especially since this behavior is central to the study. I was not absolutely clear on some points, as elaborated below:

- In the methods the authors mention the use of an automated tracking software (lines 343-344), but it's not clear to me in what way the tracking data was used, since it is not mentioned further, and later in the methods (lines 387-390; and lines 420-424) the authors describe manual scoring of approaching/listening behaviors. Was tracking data used to aid manual scoring, and if so, how?
- In the results (lines 50-53), the listening-approaching behavior is defined as follows: "The juvenile's listening approaching behavior is defined as immediate orientation and flying in close proximity toward its tutor upon hearing the tutor singing, followed by a temporary freeze of vocalizations and other ongoing behavior (e.g., feeding and preening) and strict silence (Fig. 1c-f. Supplementary Movie#1)." The movie is definitely striking! But the reader could use a bit more detail on the quantification of the behavior, namely, what is meant by "upon hearing the tutor" – how would the scorer quantify that? Also, how was the freeze of vocalizations quantified? Via video? Audio recordings (as shown in Fig 1c)? Both?
- What is meant by approaching rate (%) (Fig 1b and later), percentage of what?

2. The authors' hypothesis as to the function of the behavior they identify and describe could be more clearly explained. Some parts of the discussion read a bit vague:

- It is not absolutely clear to me whether the authors are suggesting that the approaching/listening behavior serves to improve the detail and precision of a previously acquired memory, or that it somehow facilitates the conversion of a sensory memory to a motor program for song (or both). I understand that the current study's findings cannot distinguish between these two possibilities, but it would be helpful if the authors could articulate both these hypotheses (and perhaps speculate about possible neural mechanisms of each) in the discussion. At present, they seem to interchangeably mention both possibilities without clearly distinguishing between them. In particular, the part of the discussion in lines 306-313 is unclear.
- The authors refer to an "auditory program", "intrinsic auditory program" and "intrinsic learning

program" throughout the manuscript – what does that mean exactly? Could be helpful if the authors explained early on.

Minor points:

- Lines 46-47: "we tracked each juvenile zebra finch and its social interaction with an adult tutor throughout the sensitive period of vocal learning, from 0-65 days post hatching (dph)" – when did the behavioral tracking start (probably not at day 0ph?)?
- Fig 1f: what is the purpose of the red line connecting the bird boxplots?
- Fig 1f: the legend says: "Individual birds who had earlier onset (shorter time latency after tutor sings) of approaching movement tended to have higher approaching rate (# attentive approaching movement/# tutor song)." Do we see this in the figure? If so how? If not, worth showing (maybe put the approach rate for each bird on the second y axis).
- Same comment as above for the sentence in the second paragraph of the results: "The juveniles who had a higher approaching rate also approached their tutor faster". There is no indication of statistical testing of this, and as far as I can tell, it is not shown in a figure.
- Lines 143-144: "the time-sensitive, juvenile listening-approaching behavior required previously acquired memory of tutor song". This conclusion is tentative at this point. Juveniles' approaching preferentially to their father rather than to a stranger, though suggestive, can be due to other reasons (maybe they were afraid of the stranger, maybe they prefer to approach someone they are bonded to).
- Line 147: is the reference to Fig 1b a mistake? It doesn't show whether the tutor was the father or not, or whether he was reintroduced or not. Do the authors mean that the results in 1b are from fathers that were with the kids since day 20?
- Lines 217-219: "and these juveniles were more likely to peck the tutor's beak after the tutor sang (Supplementary Movie #5)." Movie #5 shows a juvenile injected with a dopamine antagonist that does not approach. In the next paragraph the authors refer to movie #6, which seems to fit the description of Movie #5. There are only 5 movies available.
- Lines 227-229: "adult tutors reduced their singing rate in response to their less-motivated, post-injected juveniles and adult tutors were more likely to peck the beak of less motivated juveniles". Is this quantified? If so, where?
- Fig 4b: are the two panels NCM and NF? Which one is which?
- Lines 347-349: Tutor removal experiment a) – This experiment involved the removal of the pupil, rather than the tutor. This might have affected the pupil's ability/motivation to imitate in ways unrelated to the ability to socially interact with the father – maybe the juveniles were stressed – since they were away from both sibling and father, and alone in a cage?
- Lines 386-392: was manual scoring used for the dopamine group only or for all experimental birds? Why were not the dopamine group tracked?

Reviewer #1 (Remarks to the Author):

Liu et al carefully observe the movements of zebra finch clutches as juvenile birds acquire songs from tutors. They discover that juveniles tend to approach tutors right around the time they begin to sing and that this behavior peaks around the period of development that young males are primed to learn the song. The approaches happen within seconds (sometimes less) of when a tutor starts to sing, and the approaches are specific to a previously 'chosen' tutor bird. Females do not exhibit this behavior. Interestingly, approach behavior is correlated with imitation quality. These are exactly the type of careful behavioral experiments that are necessary to advance the field - they open up many future opportunities to identify neural mechanisms of tutor selection and song learning. The authors make initial inroads into mechanism by showing that activating or suppressing dopamine signaling (systemically) can influence approach behavior. They also show that immediate early gene expression in NCM, an auditory area, correlates with approach behavior. This is an important advance for the field and I support publication provided some minor issues are addressed.

Our response. Thank you for your positive and helpful evaluation to improve this manuscript.

(1) Individuals who are peak approaches approach less with DA antagonist. Individuals who are lousy approachers approach more with DA agonist. The authors do not spell out what happens when drugs were given to 'normal' birds at median levels of approach behavior - and it seems odd that they only did the agonist experiments on the low-quality approachers and the antagonist experiments on the high quality ones. Was this a post-hoc decision to specifically pull out the part of the dataset where there was a significant effect or were these the only DA drug experiments performed?

Our response: This was a post-hoc decision. Before collecting the data, we had run a number of preliminary tests to show the effect of DA agonist and antagonist in randomly selected juveniles. Our original plan was to demonstrate that DA agonist had a significant effect on enhancing a juvenile's approaching behavior, even for birds that had lower approaching rate (less motivated); and DA antagonist had a significant effect to reduce listening/approaching, even for birds with higher (more motivated) approaching rate.

The reviewer made a valid point about testing the birds with median or average level of approaching behavior. We have therefore added 3 more birds with higher approaching rate in the agonist group (n= 9 birds total) and 3 more birds with lower approaching rate to receive the administration of antagonists (n=9 birds in total), to balance out the average level

of approach rate in both experimental groups. We have revised the Methods (page 15-16), Results sections (page 8-9), Fig.3 and figure legend.

(2) Multiple sections of the paper, including the videos, refer to beak-pecking as common mode of interaction during tutoring. Tutors peck 'inattentive' juveniles and juvenile peck tutors. But this interesting behavior is only barely. In line 56 it says $r=.63$, Person's coefficient) but it is not stated in the methods what exactly was correlated with what. In addition, pecking rates and tutor singing rates are reported to be changed in the dopamine experiments but these data are not quantified.

Our response: Agree, we have revised the text and added a paragraph in Methods (page 14 of revised manuscript) to describe how we quantified the beak pecking behavior. In brief, we define "singing-induced" beak pecking as a juvenile pecking its tutor's beak immediately after (less than 5 seconds) the end of the tutor song. To quantify the singing-initiated beak pecking, the scorers viewed video recording files and then identified the number of occurrences of beak peck immediately following each tutor song within 5 seconds. We then used Pearson's correlation coefficient to test whether or not the frequency of singing-related beak pecking (number of beak-pecks/ number of tutor songs) correlates with a juvenile's tutor approaching rate (number of approaches/ number of tutor songs). The individuals who had a higher approaching rate were more likely to exhibit beak-pecking behavior. We added a supplementary figure (**Supplementary Fig. 1**, Page 3) to show this positive correlation.

We decided not to include the observation of "tutor pecked toward less motivated juveniles" in our revised manuscript because this behavior was only observed in a few tutors ($n=3$ out of 9 tutors) after their juveniles received DA antagonists. Due to the small sample size, we decided not to include these anecdotal observations in our revised manuscript.

Reviewer #2 (Remarks to the Author):

Liu et al. base their experiments on the assumption that song learning in male zebra finches involves a two-step process. A sensory period, during which a song memory is formed and a subsequent sensory-motor phase, during which the vocal output is matched with the memory in an iterative process. In the present manuscript, they hypothesize that for the second phase an active-listening behavior or the juvenile males is important and depends on prior exposure to song during the sensory phase. They therefore (1) quantify the percent of time that juvenile males approached the adult tutor during the song learning phase from 30 to 68 days after hatching. They found that peak of approaching behavior centered around 50 days of age,

coinciding with very active plastic song production of the juveniles. They also report (2) that juveniles exposed to a tutor only after 45 days of age approach the tutor, that is introduced late, much less than juveniles that are co-habiting with the tutor between 0 and 65 days of age. The juveniles exposed for longer time also develop better song copies than the juveniles exposed only after 45 days of age. (3) Using pharmacological manipulations, the dopamine agonist nomifensine and antagonist SCH23390, altered the birds 'active listening behavior'; injection of the agonist more than tripled the approach rate, whereas the antagonist-injected birds approached the tutor by a third less than saline-injected juveniles. (4) The authors also provide data on the immediate early gene *egr-1* expression in two brain regions, NCM and frontal nidopallium, which differed among groups. The authors conclude from this that the listening-approach is selective and driven by previously acquired tutor memory.

The individual findings are interesting. There is increasing evidence that social interactions between juvenile songbirds and their father/tutor and their mothers and siblings impacts song production learning in males and song preferences and mate choices in females. However, the authors did not frame their study in this context but postulate that 'active listening', e.g. flying towards the singer and suspending their own singing and other activities around 46-52 days post hatch results in better song learning, but only if a song template has already been formed earlier, before the sensory-motor phase of song learning. I find both premises hard to follow.

Firstly, Roper and Zann showed in 2006 that in zebra finches there does not seem to be a sensory 'listening only' phase preceding the 'babbling' sensory-motor phase. Both phases fully overlap, listening and rehearsing.

Our response: We respectfully disagree with Roper and Zann (2006) that the sensory learning and sensorimotor learning phases "completely overlap" at 25 dph, at least not in our finch colony and previous studies (Liu et al., 2004). Our view of the sensitive period is similar to a review by Mooney (2009) that sensory learning phase generally precedes sensorimotor learning phase with extensive overlap in both phases (see the text and Fig.2 of Mooney 2009). We agree with Gobes et al (2019) that it is difficult to draw a clear boundary about the extent of overlap between sensory and sensorimotor learning phases. It is possible that the onset of sensory learning is gradually developed after fledging (~22 day, Braaten 2010) when the fledglings can visually imprint and socially interact with their rearing tutor. And the onset of sensorimotor learning phase (production of babbling subsong) in most juvenile finches is around 30 dph (Liu et al. 2004). In our 2004 study, we carefully and intensively monitored the

onset of babbling subsong in 23 males, 7 hours per day. We showed that the onset of subsong was quite variable among individual finches, and most juveniles did not start the first subsong until around or after 30 dph (see Results in Liu et al. 2004). These results were recently confirmed in our lab by using an automatic sound-recording software to detect early subsong (a few birds didn't start their first subsong until 37-38 dph). We suspect that both genetics and the housing/ lab environments play an important role in the onset of sensorimotor learning phase (Liu et al. manuscript in submission).

It is worth noting that, in Roper and Zann 2006 paper, (1) the authors stated in Abstract "...subsong also begins on day 25suggesting that these two phases fully overlap.", but they provided no clear evidence for this conclusion. The authors did not test the onset of subsong in their experimental birds. Their evidence for the onset of subsong was based on a citation from Zann's 1996 book (which actually stated that "the first subsong begins in wild birds between 28-35 days of age...", p227). (2) although juveniles who were exposed to the tutor from 25-35 dph had significantly better tutor song matching than juveniles tutored earlier from 15-25, or 15-20 dph, early tutored birds still had up to ~40-50% tutor song matching (Fig.2. Roper and Zann 2006), suggesting some features of the tutor song had been gradually learned and the "onset" of sensory learning phase started earlier than 25 dph (see also Braaten 2010). Therefore, the conclusion of "fully overlap" at 25 day is not convincing, and it is difficult to define a clear onset of two learning phases.

Importantly, we don't see any conflicts between our study and Roper and Zann 2006 study. Our study shows that the juveniles first passively listened to the tutor song before 35 dph, but then later performed active listening behavior during the sensorimotor "plastic song" stage at around or after 46 dph, when the recognizable syllables or plastic song emerged.

To help clarify this controversial issue, we have revised our wording in Abstract and Introduction (page 1).

References:

- Braaten, R.F., 2010. Song recognition in zebra finches: are there sensitive periods for song memorization? *Learn. Motiv.* 41, 202–212.
- Gobes SMH, Jennings RB, Maeda RK (2019) The sensitive period for auditory-vocal learning in the zebra finch: Consequences of limited-model availability and multiple-tutor paradigms on song imitation. *Behavioural processes* 163:5-12.
- Liu WC, Gardner TJ, Nottebohm F (2004) Juvenile zebra finches can use multiple strategies to learn the same song. *Proceedings of the National Academy of Sciences of the United States of America* 101:18177-18182.
- Mooney R (2009) Neural mechanisms for learned birdsong. *Learning & memory* 16:655-669.
- Zann, R.A. (1996) *The Zebra Finch*. Oxford University Press.

Second, the data of approaching the singer and beak pecking are hard to evaluate without knowing how often a juvenile approaches the tutor/pecks his beak outside of the song context. Birds fly/hop around constantly and do a lot of beak pecking.

Our response: We want to clarify that “tutor singing-induced” beak pecking is not our focal hypothesis to be tested and it is not one of the main characteristics of listening/ approaching behavior (please see our definition of listening/approaching behavior in page 2). Many juveniles did not have beak pecking behavior when they performed listening/ approaching after tutor singing. However, we did notice that juveniles that exhibited a higher approaching rate (number of approaches/number of tutor songs), there juveniles were more likely beak-pecking their tutors immediately after tutor singing (within 5 seconds). We only show there was such a correlation in this study. We have also added a supplementary figure to show this correlation (**Supplementary figure 1**, page 3). We will explore the function and mechanism of “singing-induced” beak pecking behavior in a follow-up study.

In order to know whether juveniles approached tutors more as a result of the tutor singing I would think that it is necessary to know how often juveniles approached tutors during the times that tutors were not singing. Clearly, approaching and peak pecking could serve multiple behavioral functions, so if there was not difference between tutor-singing and tutor-silent it would not devalue the data but if there was a difference it would certainly help to convince me that the approaching and beak pecking were related to song learning. If there were no difference, it could still be that birds approach the tutor for different reasons, when he sings to listen and when he does not sing for other reasons or pure chance. But it is hard to evaluate the reasons why a juvenile flies towards the tutor without more information on chance/different reasons.

Our response: To address Reviewer #2’s comments, we have provided additional data analysis and evidence (see **Supplementary Figure 3**, page 5) that overall (including non-singing and singing related) approaching between an approaching juvenile and its tutor did not correlate with tutor song imitation and did not correlate with “tutor singing-induced” listening/ approaching behavior. We measured how often (frequency) and how long (duration) a juvenile approached its tutor under all social contexts. We used a movement tracking software (Ethovision 14, see Methods, page 14) to automatically track the frequency and duration of close approaching between an approaching actor (juveniles) and its receiver (tutors) from 30-65 dph. We set up the threshold between the approaching actor and the receiver as within 5 cm of distance. The software can automatically quantify **1**) the frequency

of juvenile approaching tutor within 5 cm of distance (that is, number of approaching per 6 minute recording); and **2**) mean cumulative duration of approaching (percentage of time when two birds stay within 5 cm of distance, per 6 minute recording) from 30-65 dph. Totally 17 males that had complete video recordings were used in this analysis. In our preliminary test, this automatic quantification by the tracking software was confirmed by 3 student scorers' manual scoring in blind.

Our results show there was no correlation between “tutor singing-induced” approaching rate and overall (non-singing) approaching rate under all social conditions. And there was no correlation between similarity match to the tutor song and overall approaching (frequency and duration) from a juvenile to its tutor. We have added these results in **Supplementary Fig. 3**.

Because this automatic tracking software does not detect sounds, this automatic tracking is not able to distinguish singing or non-singing related approaching movement. We thus used this software to quantify the overall approaching behavior, and “tutor singing-initiated” listening/approaching behavior had to be quantified manually by our scorers.

We have revised the main text in Results (pages 4-5, line 126), and added two paragraphs in Methods to describe how we quantified the overall approaching rate (page 13).

The fact that the cages (in the videos) seem to be somewhat small and opportunities for sitting are limited does not aid in this evaluation. A larger cage with more places to sit would make it easier to evaluate whether an approach towards the tutor was deliberate or chance. The fact that birds suspend vocalizations for around a minute after they join the tutor is consistent with the notion that they are listening, but one would also like to have more information on whether and when juveniles suspend vocalizations for a minute outside of the ‘listening context’. If they never do this except when ‘listening’ to the tutor singing, this would be more convincing.

Our response: We respectfully disagree with the Reviewer#2 that the cage size is small, which may affect the behavioral quantification. **1**) When we quantified the singing-related approaching behavior using this cage setup, four student scorers independently rated the approaching behavior and our scorers almost unanimously agreed (>93%) what was singing-initiated approaching, attentive but no approaching, or approaching under other contexts. It is worth noting that our student scorers did not have knowledge of zebra finches and their age/gender differences in morphology. They identified each individual bird based on the color

paints. **2)** As mentioned above, the cage setup allowed us to use an automatic tracking software (Ethovision XT14) to track the movement from an approaching juvenile (actor) to its tutor (receiver) under all social conditions. And in our preliminary test, the approaching data analyzed by the automatic tracking software were later confirmed by 3 student scorers' manual scoring of approaching behavior under all conditions. Therefore we are confident about our analysis and cage size.

Our cage has a dimension of 35 x 52 x 61cm, and each perch length is 35 cm, which allowed us to reliably quantify the approaching distance of less than 5cm. Based on IACUC recommendation, this cage space can normally hold 10 adult birds. In this study, we kept only 3 or 4 birds per cage, in which the birds have a lot of space to move around and allowed us to quantify the distance of interacting birds. Before collecting our data, we had run preliminary tests using different cage sizes and analyzed how the software can capture and quantify each individual movement, and this analysis was later confirmed by our scorers to manually quantify the listening/approaching behavior from video recordings. We are confident that our quantification of approaching behavior is robust. The video files will be deposited for open access, and we welcome Reviewer#2 to view and quantify the video recordings.

To address the reviewer's comments, we have revised the Methods sections and provided more detailed information about cage setup and how we quantified the approaching behavior (see pages 13-14, line 401, in the revised ms).

In addition, many of the experiments have other confounds. For instance, introduction of an unfamiliar adult male does not only change the behavior of the juvenile because he might not have been imprinted on him in an earlier phase but the unfamiliar male might also behave more aggressive towards the unfamiliar juvenile. Likewise, exposing a juvenile to a tutor only at day 45 not only prevents the early imprinting phase but also changes the total amount of song exposure. The tutor who has not been exposed to his son for the last 35 days might also have a substantially different behavior towards his son that had they cohabited the entire time.

Our response: We have added more experiments to address the reviewer #2's comments (see below for details).

Finally, immediate early gene expression, at least in NCM, might be affected by the total amount of song heard in the different groups, which was not quantified as far as I could gather.

Our response: Agree. We have provided additional analysis to quantify the NCM based on the song exposure, see the revised Methods (page 17, line 580) and **Fig. 4** (page 10).

In summary, the different findings are each interesting and some experiments involved a large number of birds, and groups often consisted of sibling cohorts to control for genetic effects, which is very nice. However, the premise of paper (two phases of song learning) is not substantiated by previous data and moreover the story that the authors construct around the individual behaviors does not hang sufficiently well together, given that alternate explanations are possible for each experiment. The manuscript would benefit from major revisions, see specific points below.

Our response: We have substantially revised the manuscript and added more data to clarify the reviewers' comments (see below for details).

Line 24 „Vocal learning in songbirds is generally considered a two-step learning process (Fig. 1a).“ Vocal learning in songbirds follows very different trajectories depending on the species (see Supplemental Info Box in the cited article Bolhuis et al 2010).

Our response: Please see our earlier reply. Our view of the sensitive period in songbirds (and in zebra finches) is similar to a review article by Mooney (2009). We cited the review article of Bolhuis et al 2010 to show different views on this topic.

The time line of the x-axis in Fig 1a suggests zebra finches, but in zebra finches sensory and sensory-motor phases overlap completely (Roper and Zann, 2006).

Our response: We respectfully disagree with the conclusion from Roper and Zann 2006, at least not in our zebra finch colony, and not in our previous studies (Liu et al 2004). Please see our earlier reply.

Line 29: ‚restore’ should be ‚store’?

Our response: We have corrected the wording.

Line 32: „Moreover, the growing number of field studies suggest that, even if model memory is first acquired (my italics) during the sensory learning phase, later re-exposure of the model during the sensorimotor learning phase is critical to ensure selective attention on final song commitment and precise model imitation during territory establishment (8-10).“ ‘Acquired’ sounds very much like ‘learned’ but Reference 9 (Liu and Nottebohm 2007) shows clearly that the ‘precursor’ songs of chipping sparrows are not acquired during a sensory learning phase with reference to an external model. Instead, it is the later exposure to a

model/neighbor that causes one of the precursor songs to be modified to the final version.
Rephrase to be clearer.

Our response: We have rephrased the sentences and deleted the citation of Liu and Nottebohm 2007 to be clearer (page 1, lines 32-36).

We cited Liu and Nottebohm 2007 paper to show the importance of tutor song exposure during the sensorimotor “plastic song” stage, and this learning strategy has a limitation. As the leading author in that paper, we (WL) showed that, without early song exposure, late hatching juvenile chipping sparrows CAN imitate songs from the tutor during later exposure of tutor song (i.e., the sensorimotor “plastic song” stage), BUT this strategy has a limitation-- not all the juveniles can imitate, only the juveniles whose limited “precursor syllables” that matched the tutor song. Actually 3 out of 7 birds completely failed to imitate the tutor (see Results in Liu and Nottebohm, 2007). We should have made it clear when citing this paper.

More common or reliable song imitation of chipping sparrows in nature happens when juvenile sparrows were first exposed and acquired tutor song in the hatching year (early sensory learning phase) and then re-exposed to the same tutor song during the sensorimotor “plastic song” stage, as our field study suggested (Liu and Kroodsma, 2006). We have added the paper by Liu and Kroodsma (2006) in our revised ms to clarify our point.

Importantly, the growing number of field studies in a number of seasonal songbird species (field sparrows, savannah sparrows, and white-crowned sparrows; Nelson, 1992, 2000, Mennill, 2018) support the idea that song is more commonly and more precisely imitated when tutor song is first exposed during the hatching year (early sensory learning phase) and then re-exposed during the following spring (sensorimotor, plastic song stage).

References.

- Liu WC, Kroodsma DE (2006) Song learning by Chipping Sparrows: When, where, and from whom. *Condor* 108:509-517.
- Liu WC, Nottebohm F (2007) A learning program that ensures prompt and versatile vocal imitation. . *Proceedings of the National Academy of Sciences of the United States of America* 104: 20398-20403.
- Mennill DJea (2018) Wild Birds Learn Songs from Experimental Vocal Tutors. *Current biology* 28:3273-3278 e3274.
- Nelson DA (1992) Song overproduction and selective attrition during song development in the field sparrow (*Spizella pusilla*). *Behavior, Ecology, and Sociobiology* 30:415-424.
- Nelson DA (2000) Song overproduction, selective attrition and song dialects in the white-crowned sparrow. *Anim Behav* 60:887-898.

Line 69; “peaks” should be past tense “peaked”

Our response: Thanks, we have corrected it in our revised manuscript.

Line 79: legend to figure 1f) missing info what the red line depicts

Our response: The red line depicts the spline regression line between two variables. We decided to delete the red line in our revised ms (Fig. 1f, page2), because the line did not convey the main point of this figure. We used Pearson's correlation coefficient to test the correlation between individual's approaching rate and onset (time latency) of approaching behavior.

Line 90: "The juveniles who had a higher approaching rate also approached their tutor faster (i.e., shorter time latency between the onset of tutor song and tutor- approaching, Fig. 1f)."
This sentence implies that there is a correlation between approaching rate and latency, but Fig 1f does not show this.

Our response: Agree. We have added each individual's approaching rate (number of approaches/number of tutor song) on the X axis of Fig. 1f, and we have revised the figure legend (page 3) to show the correlation between the approach rate and latency.

Line 96 "sit up gesture' (Supplementary Movie# 4)."
Please indicate in the movie when the ,sit-up gesture' starts and ends. Alternatively, provide a series of still photos. I am not sure when the sit-up is happening.

Our response: We decided to delete the wording "sit up gesture" in Supplementary Movie #4, To address the Reviewer#2's comments, we did a more comprehensive quantitative analysis and realized the adult tutors seem to have a diverse and individual-specific singing gestures before song, and not all of the adult tutors had sit up gesture before song. Because of this individual variability, we decided to delete the wording "sit up gesture", and we will explore the tutor's singing gesture in a follow-up study.

Line 124 Supplementary Figure 2. „There was no correlation between an individual's approaching rate and its proximity to the tutor“
How was proximity quantified? What does the % on the y axis refer to? Percent time of how many video minutes quantified? And what was the distance definition to qualify as ,proximity'?

Our response: We have substantially revised the text in the Methods section (adding a paragraph on pages 13, lines 419-433 of revised ms) to better define the proximity and describe how we quantify the proximity.

As mentioned earlier, proximity is defined as the close distance of less than 5 cm between an approaching juvenile and its tutor under all social contexts, tutor singing or not. We used a movement tracking software, Ethovision XT14, to automatically track the occurrence of approaching proximity between an approaching actor (juveniles) and a receiver (tutors). We set up the proximity threshold between the approaching actor and the receiver as within a distance of 5 cm. The software automatically quantified **1**) the frequency of juvenile approaching tutor (number of approaches from a juvenile to its tutor within 5 cm of distance, per 6 minute video recording); and **2**) mean cumulative duration of approaching (percentage of time period when two birds stay within 5 cm of distance, per 6 minute recording) during the sensitive period of song learning (30-65 dph). Totally 17 males that had complete video recording were used in this analysis.

In **Supplementary Fig 3**, “% on the y axis” refers to the percentage of cumulative time period when an approaching juvenile stayed with its tutor within 5 cm of distance per 6 minute recordings, recorded from 30-65 dph. We have added this information in Methods, page 14).

In our revised manuscript, we have added two more figures and figure legend in revised **Supplementary Fig.3** to show that **1**) the overall approaching rate is not correlated with singing-initiated approaching behavior; **2**) a juvenile approaching its tutor under all social conditions is not associated with tutor song imitation capability.

Line 146 „males performed robust approaching and listening behavior at 45-55 dph when they were re-exposed or repeatedly exposed to the same father tutor (Fig. 1b)“ Fig 1b does not show data about re-exposure.

Our response: Agree, we have deleted the wording “re-exposed” in our revised ms (page 6, line178). The results for “re-exposure experiments” are presented in line180 and Fig.2a.

Line 159 „The results suggested that highly motivated approaching behavior is driven by re-exposed or repeated exposed tutor song memory.“ I don't think that can be concluded, since there are many other reasons in addition to number of approaches that could cause juveniles that had exposure to their father only after 45 have poor song copying success (less time with the tutor, different social relationship with the tutor, sensitive period no longer optimal, siblings already copying each other during the time without tutor etc)

Our response: Our focal hypothesis and conclusion sentence in that paragraph is “**highly motivated approaching behavior may require previously acquired tutor memory**”. Our

focal hypothesis was NOT aimed to test “number of approaching to cause song copy success”. We conclude juvenile listening/approaching behavior may require previously acquired tutor song memory, based on the following evidence. **1)** If a juvenile was tutored by its father or foster tutor from 0-65 or 20-65 dph, juveniles performed robust approaching/listening behavior at 44-55 dph when they were continuously exposed to the same adult tutor (Fig. 1b; n=33 birds). **2)** If the father tutor was temporarily removed from 35-45 dph (n=8 birds), juvenile males performed significantly higher approaching and listening behavior at 45-55 dph when they were re-exposed to the same father tutor (Fig. 2a). **3)** If a juvenile was not exposed to a tutor during the early sensory period (i.e., tutor removal between 10-45 dph), juveniles significantly reduced tutor-approaching behavior after the father tutor was later-introduced at 46 dph (Fig. 2a). These results are consistent with a previous study by Chen et al., (2016) that juveniles exhibited attentive but no approaching behavior when they were not exposed to a social tutor until after 40 dph. **4)** If the previously exposed father tutor and an adult male stranger were both presented with the juveniles after 45 dph, juveniles unanimously approached toward the father tutor’s singing. However, as the reviewer pointed out, this could be due to the social bonding with the father, or fear of the stranger. **5)** To rule out the possible social confound, as Reviewer#2 suggested, we provided additional song playback experiments to test our hypothesis. Juveniles previously exposed to the tutor until 45 dph had significantly more approaching behaviors toward the speakers of playback of the same tutor song after tutor removal at 46 dph, compared to the playback of a stranger song, as juveniles rarely approached the speakers of stranger song playback. **6)** These experiments were also supported by *Egr1* expression study (**Fig.4**).

Overall, these results show that the attentive listening/approaching behavior (46-55 dph) occurred when juveniles had continuous exposure or re-exposure of live, adult tutors (continuous song playback from 30-65 dph, however, induced much less and weaker approaching behavior, **Fig. 2a**). Continuous exposure or re-exposure of the same tutor, of course, may lead to more tutor/song exposure to strengthen the juvenile’s song/tutor memory. This is consistent with our conclusion “highly motivated approaching behavior may require previously acquired tutor song memory”

We have substantially revised the paragraph to make it clear (page 6, lines173-195).

Chen Y, Matheson LE, Sakata JT (2016) Mechanisms underlying the social enhancement of vocal learning in songbirds. *Proceedings of the National Academy of Sciences of the United States of America* 113:6641-6646.

Line 177 Figure 2 c and d: Given that the 0-65 day group had 20 days more exposure to tutor song than the 0-45 day group it is hard to know whether the lack of interaction with the father or the lack of exposure caused the lower copy success. The 0-45 day group could hear the tutor from 2m away but not see him. One could argue that this acoustic distance and visual isolation might cause the juveniles to categorize the song as no longer socially relevant. This experiment would need to be better controlled in terms of potential variables affecting tutor imitation.

Our response: We agree that this experimental design did not control potential social confounds. We have thus decided to delete this experiment and the results (page6, lines197-209). Nonetheless, these results are consistent with a previous study by Eales (1985) when juveniles were housed together with their father tutors until 35, 50, or 65 dph. The juveniles who learned best, and copied most song syllables were the ones tutored until 65 dph.

We believe that we have provided robust results to support the main point of this paragraph (page 6) that “The attentive listening/approaching behavior is associated with vocal imitation capability”. **1)** Juvenile males (with continuous tutor exposure from 0-65 or 20-65 dph) that had a higher rate of the attentive approaching to their tutors also developed better tutor imitation in their crystallized song (Fig. 2b, n=23 birds). **2)** Juveniles who were continuously exposed or re-exposed to the social tutors until 65 dph had higher approaching rate (Fig. 2a), and also had a significant better song imitation compared to the birds were tutored after 45 dph, or with tutor playback (Fig. 2c-d).

Eales, L. A. Song learning in zebra finches: some effects of song model availability on what is learnt and when. *Anim Behav* **33**, 1293-1300 (1985).

Line 240 „Each red dot is one single recording of the approaching movement.“ This means that some birds might have contributed many more data than others to plots b and d. I suppose this is for illustrative purposes only and the statistics were done on the averaged data of n=6 birds? If the authors want to keep the graph in panel b and d as is, then I suggest to mark the individual birds with different symbols or colors so that the identity of the birds can at least be eyeballed visually.

Our response: In our revised ms, we have added a supplementary figure to address Reviewer #2’s request. The figure shows each individual’s approaching behavior with different colors before and after agonist injection (see **Supplementary Fig.4**, page 9).

Line 251 „The expression level was significantly higher in juveniles who approach“ approach should be past tense.

Our response: We have corrected that (Line 282, page 9).

Line 262 Figure 4: (b) top and bottom panel not specified which is NCM and which is NF.

Our response: We have added a label of NCM and NF in the revised **Fig. 4** (page 10).

Line 328 animals. A table with the animals used in the different experiments would be useful. As it is it is hard to have to go back and forth to find this information, partly in the legends, partly in the text and partly in the methods.

Our response: Agree, we added a supplementary table to show the number of animals used in each experiment (please see **Supplementary Table 1** in page 12 of revised ms).

Line 392: is there a reason why the authors used events recording rather than duration (“the number of times each bird approached, number of times each bird attended, and number of times the father sang”)

Our response: The quantitative measure of approaching rate and latency allowed us to test our hypothesis that attentive approaching behavior is associated with dopamine. The quantification of approaching rate in response to tutor singing (number of approaches/ number of tutor songs) is one of our main quantitative measures throughout our manuscript. We showed that approaching rate and the onset of approaching (latency) had a great individual variability (**Fig. 1b-1f, Fig. 2b.** and **Supplementary Fig.2**), and this individual variability was associated with precision of vocal imitation, In the dopamine agonist/ antagonist experiment, we suspect that individuals with a higher approaching rate and shorter time latency that we observed in Fig. 1 might be associated with a greater motivation state than individuals with less attentive or no approaching, and this motivated behavior may be regulated by dopamine. We thus used approaching rate and latency (onset of approaching) as a quantitative measure of dopamine-regulated behavior.

We have revised the Methods (page 15, Lines 503-504).

+ to make it clear.

Line 398: what age was the agonist injected? Seems like at a different age than the antagonist. Please state. Possible confound. Injecting unresponsive same age birds with agonist and responsive birds with antagonists would be better, and presenting data as ‘before-after’ by bird rather than group.

Our response: They were injected at the same age. As mentioned in Methods, we injected agonist and antagonist at approximately the same age (between 46 to 55 dph), during the

peak of the listening and approaching behavior. We injected antagonists when juveniles showed a higher approaching rate, and injected agonists when birds showed lower approaching rates at around the same age. We have made it clear in our revised manuscript (page 15, Line 509).

Additionally, to address Reviewer#1's comments (see our reply above), we have added additional 6 birds (3 for agonists, 3 for antagonists at the same age group) for this analysis.

Line 407 ff: it would be nice to see a time line of the assayed behavior for the 1 hour before injection and then three hours post-injection, in addition to the data provided in Fig 3, to appreciate the within animal effect of the drug taking effect and the effect disappearing again.

Our response: We added a supplementary figure that shows the timeline for the 1 hour before injection and 2 hours post-injection (**Supplementary Fig. 4b** in page 9 of revised ms).

Line 455 in situ hybridization paragraph: missing a lot of information. As previously described has no reference, borders of NCM, NF and MFV not explained, how many sections were quantified and how, normalization of one group against averages of two other groups unclear how and why. MFV – the abbreviation is not explained and the results for MFV do not appear in the manuscript.

Our response:

1. We have provided two references about our in-situ hybridization protocols. We followed the previous protocols from Liu and Nottebohm (2005) and Hayase et al. (2018), we have added more detailed information in revised Methods (page 16).
2. For quantification of NCM, three sections (per hemisphere) at the medial parasagittal position (between 180-330 μm from the midline) were collected. Ventricle was used as dorsal, caudal, and ventral boundaries. For NF, three sections from parasagittal position (1450 μm from the midline) were collected. LaM (Lamina mesopallialis) was used as dorsal, medial, and lateral boundaries, and Bas (Nucleus basorostralis pallii) as ventrocaudal boundary. All data were first normalized to the mean level of *Egr1* expression in silent condition at the same age (page 17 of revised ms)
3. We decided not to include MFV because we didn't see a consistent and significant difference in *Egr1* expression among different experimental conditions.

References.

Hayase S, Wang H, Ohgushi E, Kobayashi M, Mori C, Horita H, Mineta K, Liu WC, Wada K (2018) Vocal practice regulates singing activity-dependent genes underlying age-independent vocal learning in songbirds. *PLoS biology* 16:e2006537.

Liu WC, Nottebohm F (2005) Variable rate of singing and variable song duration are associated with high immediate early gene expression in two anterior forebrain song nuclei. *Proceedings of the National Academy of Sciences of the United States of America* 102:10724-10729.

Reviewer #3 (Remarks to the Author):

In this study, Liu et al examine the sensory phase of vocal learning in zebra finches, and show that it is composed of two distinct stages: a stage of passive listening and formation of a memory of the learning target, and a stage of active listening and behavioral engagement of a juvenile with an adult “tutor”. The authors demonstrate that the latter stage involves approaching and listening behaviors that are directed specifically toward the tutor whose song was heard during the first (passive) stage of learning; that it facilitates vocal imitation; and that it is modulated by dopamine and accompanied by activation of a high auditory forebrain region thought to store auditory memories. The authors hypothesize that attentive and active listening to a previously memorized song serves to fine tune the stored memory of a complex song and thus facilitate precise imitation. The findings are novel and extremely interesting, beautifully showing how juveniles kept with their fathers in semi-natural conditions immediately cease any on-going activities (including their own singing) and approach their father as he starts singing, and sometimes even right before he starts singing (indicating that juveniles are highly attentive to any behaviors that may indicate an intention to sing in the father). The results are robust and convincing. The experimental and analysis methods are adequate (though need some elaboration, see below), and overall the writing and figure presentation are clear and to the point. The study can therefore be a valuable contribution to *Communications Biology*. I have two main comments that should be easy to address:

Our response: Thank you for your positive and helpful evaluation to improve this manuscript.

1. The method for quantifying the approaching/listening behavior should be described in more detail, both in the methods section and in the results, especially since this behavior is central to the study. I was not absolutely clear on some points, as elaborated below:

1). In the methods the authors mention the use of an automated tracking software (lines 343-344), but it’s not clear to me in what way the tracking data was used, since it is not mentioned further, and later in the methods (lines 387-390; and lines 420-424) the authors describe

manual scoring of approaching/listening behaviors. Was tracking data used to aid manual scoring, and if so, how?

Our response: Agree. We have added a few more paragraphs to clarify the quantification of approaching/ listening behavior and proximity. In brief, we first used a wi-fi network camera to record the behavior from 0800-1000 every other day during 30-65 dph. Then from each video recording, four scorers manually scored approaching/listening behaviors. For the initial screening, scorers used approximately 10 cm (and then 5 cm) of distance between a juvenile and its tutor as a threshold before further quantification. The quantification of listening/approaching behavior was done manually, and a tracking software (EthoVision XT14, Noldus) was used to aid manual scoring. This software allowed us to visualize and measure the approximate distance (less than 5 cm) between the tutor (receiver) and the approaching juvenile (actor). The tracking software also allowed us to slow down the video frames and closely quantify the approximate timing (onset and duration) of approaching behavior, and the duration of vocal silence or freezing behavior.

We also used a tracking software for automatic scoring the overall approaching between an approaching juvenile and its tutor under all social contexts (including singing-initiated or non-singing related approaching behavior). The software allowed us to automatically measure the frequency (mean number of approaches within 5 cm, per 6 minute recording) and duration of proximity (percentage of cumulative time period when a juvenile stayed together with its tutor <5 cm of distance, per 6 minute recording) between an approaching juvenile and its tutor under all social conditions (**Supplementary Fig. 3**). The close proximity threshold was set up as the distance of <5 cm between the actor (approaching juvenile) and the receiver (tutor).

Because this automatic scoring software does not detect sounds, it does not distinguish “tutor-singing initiated” approaching behavior from other non-singing related approaching behavior. We thus had to use manual scoring to quantify “tutor singing-initiated” listening/approaching behavior,

We have added three paragraphs to better describe our methods of quantification (pages 13-14 of revised ms).

2) In the results (lines 50-53), the listening-approaching behavior is defined as follows: “The juvenile’s listening approaching behavior is defined as immediate orientation and flying in close proximity toward its tutor upon hearing the tutor singing, followed by a temporary freeze

of vocalizations and other ongoing behavior (e.g., feeding and preening) and strict silence (Fig. 1c-f. Supplementary Movie#1).” The movie is definitely striking! But the reader could use a bit more detail on the quantification of the behavior, namely, what is meant by “upon hearing the tutor” – how would the scorer quantify that? Also, how was the freeze of vocalizations quantified? Via video? Audio recordings (as shown in Fig 1c)? Both?

Our response: Agree. We have revised the text (page 2, lines 49-52, of the revised ms) and provided a clear description of the behavior. In brief, approaching behavior is defined as immediate orientation and flying in close proximity (approximately less than 5 cm between tutor and juvenile) toward its tutor upon hearing the tutor singing (that is, within 5 seconds after the onset of tutor singing), followed by a temporary freeze of ongoing movements (vocalizations and other behavior) for at least 1 second.

Four scorers manually scored the behavior by viewing the video recordings, please see our previous reply in **1**). The video recording files come with audio sounds, and as mentioned earlier, we used a tracking software to slow down the video recordings and manually examine the close distance between the juvenile and the tutor, the onset timing or duration of the approaching or freezing behavior.

- What is meant by approaching rate (%) (Fig 1b and later), percentage of what?

Our response: We have deleted the “percentage” of the approaching rate to make our figure presentation more consistent. The approaching rate is defined as “the number of approaches to the tutor/ the number of tutor songs”. In a few figures, we used percentage of approaching rate (that is, approaching rate x 100%), but in other figures, we did not use the percentage. We thus decided to delete the calculation of “percentage” to be more consistent. We have rephrased the approaching rate in all the figures and figure legends.

2. The authors’ hypothesis as to the function of the behavior they identify and describe could be more clearly explained. Some parts of the discussion read a bit vague:

- It is not absolutely clear to me whether the authors are suggesting that the approaching/listening behavior serves to improve the detail and precision of a previously acquired memory, or that it somehow facilitates the conversion of a sensory memory to a motor program for song (or both). I understand that the current study’s findings cannot distinguish between these two possibilities, but it would be helpful if the authors could articulate both these hypotheses (and perhaps speculate about possible neural mechanisms of each) in the discussion. At present, they seem to interchangeably mention both

possibilities without clearly distinguishing between them. In particular, the part of the discussion in lines 306-313 is unclear.

Our response: We agree and have revised the Discussion section (page 11, lines 348-355). In brief, we speculate that the active and attentive approaching/listening behavior by juvenile zebra finches may serve one and/or two possible functions in the developing vocal learning circuits: **1)** attentive approaching may activate or strengthen the previously acquired tutor memory that allows more precise vocal imitation; and we speculate the listening/approaching behavior may strengthen forebrain auditory memory system (Gobes and Bolhuis, 2007; Bolhuis and Moorman, 2015). **2)** attentive approaching may selectively facilitate the conversion of a previously acquired auditory memory to vocal motor output. The song circuit for sensorimotor integration, including cortical-basal ganglia dopaminergic pathways, may be strengthened (Tanaka et al. 2018; Fee and Goldberg, 2011).

Gobes, S. M. & Bolhuis, J. J. Birdsong memory: a neural dissociation between song recognition and production. *Current Biology* **17**, 789-793 (2007).

Bolhuis, J. J. & Moorman, S. Birdsong memory and the brain: in search of the template. *Neuroscience and biobehavioral reviews* **50**, 41-55, doi:10.1016/j.neubiorev.2014.11.019 (2015).

Tanaka, M., Sun, F., Li, Y. & Mooney, R. A mesocortical dopamine circuit enables the cultural transmission of vocal behaviour. *Nature* **563**, 117-120, doi:10.1038/s41586-018-0636-7 (2018).

Fee, M. S. & Goldberg, J. H. A hypothesis for basal ganglia-dependent reinforcement learning in the songbird. *Neuroscience* **198**, 152-170 (2011).

• The authors refer to an “auditory program”, “intrinsic auditory program” and “intrinsic learning program” throughout the manuscript – what does that mean exactly? Could be helpful if the authors explained early on.

Our response: In the revised ms (Line 40, Page 1), we explained “auditory program” earlier in the Introduction section. We use an auditory program to emphasize a juvenile's active listening behavior that is universal and selectively attend to previously memorized tutor song at a specific time window.

We decided to delete the wording “intrinsic”. We used the word “intrinsic” to explain the listening/approaching behavior is universal in juvenile zebra finches during the sensitive period of song development, and it seems to be internally motivated behavior. However, we do not have good evidence to suggest that this behavior is due to internally rewarded motivation.

Minor points:

- Lines 46-47: “we tracked each juvenile zebra finch and its social interaction with an adult tutor throughout the sensitive period of vocal learning, from 0-65 days post hatching (dph)” – when did the behavioral tracking start (probably not at day 0ph?)?

Our response: We have corrected this error in the revised ms (Page 2, Lines 46-47), we tracked the behavior from 30-65 dph.

- Fig 1f: what is the purpose of the red line connecting the bird boxplots?

Our response: The red line depicts the spline regression line between two variables. We decided to delete the red line in our revised ms, because the line does not convey the main point of the figure. We used Pearson’s correlation coefficient to test the correlation between individual approaching rate and latency of approaching behavior (Fig.1f of revised ms, Page 2).

- Fig 1f: the legend says: “Individual birds who had earlier onset (shorter time latency after tutor sings) of approaching movement tended to have higher approaching rate (# attentive approaching movement/# tutor song).” Do we see this in the figure? If so how? If not, worth showing (maybe put the approach rate for each bird on the second y axis.

- Same comment as above for the sentence in the second paragraph of the results: “The juveniles who had a higher approaching rate also approached their tutor faster”. There is no indication of statistical testing of this, and as far as I can tell, it is not shown in a figure.

Our response: Agree, we have revised Fig. 1f and added statistics (Page 3, Lines 80-85). We have revised **Fig. 1f**, and added the approaching rate of each bird on the X axis.

- Lines 143-144: “the time-sensitive, juvenile listening-approaching behavior required previously acquired memory of tutor song”. This conclusion is tentative at this point. Juveniles’ approaching preferentially to their father rather than to a stranger, though suggestive, can be due to other reasons (maybe they were afraid of the stranger, maybe they prefer to approach someone they are bonded to).

Our response: To address the reviewer’s comments, we have added a song playback experiment (see below) in the Methods (Page 14, Lines 471-482) and Results sections (page 6, Lines 189-195). And we have tuned down the wording (see the revised text).

We conclude “juvenile listening/approaching behavior may require previously acquired tutor song memory” based on the following evidence. **1)** If a juvenile was tutored by its father or foster tutor from 0-65 or 20-65 dph, juveniles performed robust approaching/listening

behavior at 44-55 dph when they were continuously exposed to the same adult tutor (**Fig. 1b**; n=33 birds). **2)** If the father tutor was temporarily removed from 35-45 dph (n=8 birds), juvenile males performed significantly higher approaching and listening behavior at 45-55 dph when they were re-exposed to the same father tutor (**Fig. 2a**). **3)** If a juvenile was not exposed to a tutor during the early sensory period (i.e., tutor removal between 10-45 dph), juveniles significantly reduced tutor-approaching behavior after the father tutor was later introduced at 46 dph (**Fig. 2a**). These results are consistent with a previous study by Chen et al., (2016) that juveniles exhibited attentive but not approaching behavior when they were not exposed to a social tutor until 40 dph. **4)** If the father tutor and an adult male stranger were both presented with the juveniles after 45 dph, juveniles unanimously approached to the father tutor's singing. However, as the reviewer pointed out, this could be due to the closer social bonding with the father, or fear of the stranger. **5)** To rule out the possible social confound, we provided additional song playback experiments to test our hypothesis. Juveniles previously exposed to the tutor until 45 dph had significantly more approaching behaviors to the speakers of playback of the same tutor song after tutor removal at 46 dph, compared to the playback of a stranger song, as juveniles rarely approached the speakers of stranger song playback. **6)** These experiments were also supported by *Egr1* expression study (**Fig.4**) that juveniles without early tutor exposure had less approaching rate and lower *Egr1* expression in NCM and NF.

• Line 147: is the reference to Fig 1b a mistake? It doesn't show whether the tutor was the father or not, or whether he was reintroduced or not. Do the authors mean that the results in 1b are from fathers that were with the kids since day 20?

Our response: Thanks for pointing out the mistake. **Fig.1b** is a reference to show that juveniles were continuously exposed to the same tutor from 20-65 dph. The results of tutor re-exposure were presented in **Fig. 2a** (and Page 6, Lines 179-181). We have rephrased the sentence in Results (page 6) and revised **Fig. 2a**.

• Lines 217-219: "and these juveniles were more likely to peck the tutor's beak after the tutor sang (Supplementary Movie #5)." Movie #5 shows a juvenile injected with a dopamine antagonist that does not approach. In the next paragraph the authors refer to movie #6, which seems to fit the description of Movie #5. There are only 5 movies available.

Our response: We have corrected the mistake, there are only 4 supplementary movies in our revised manuscript.

• Lines 227-229: “adult tutors reduced their singing rate in response to their less-motivated, post-injected juveniles and adult tutors were more likely to peck the beak of less motivated juveniles”. Is this quantified? If so, where?

Our response: Due to the small sample size (n=3 out of 9 tutors showed beak pecking to the post-injected young), we were not able to quantify this anecdotal behavioral observation and run statistics. We decided NOT to include the sentence that “beak pecking from tutor to the less motivated juveniles” in our revised manuscript (Lines 245-252). We will leave it for the future study.

• Fig 4b: are the two panels NCM and NF? Which one is which?

Our response: We have added the labels on NCM and NF in the revised **Fig. 4b**.

• Lines 347-349: Tutor removal experiment a) – This experiment involved the removal of the pupil, rather than the tutor. This might have affected the pupil’s ability/motivation to imitate in ways unrelated to the ability to socially interact with the father – maybe the juveniles were stressed – since they were away from both sibling and father, and alone in a cage?

Our response: We agree that this experimental design did not control potential social confounds. We have thus decided to delete this experiment and the results (Page 6). Nonetheless, these results are consistent with a previous study by Eales (1985) when juveniles were housed together with their father tutors until 35, 50, or 65 dph before isolation. The juveniles who learned best, and copied most song syllables were the ones tutored until 65 dph.

We believe that we have provided enough evidence to support our conclusion that “the attentive listening/approaching behavior is associated with vocal imitation capability”. **1)** Individual juvenile (with continuous tutor exposure from 0-65 or 20-65 dph) who had a higher rate of the attentive approaching to their tutors also developed better tutor imitation in their crystallized song (n=23 birds, Fig. 2b). **2)** Juveniles with continuous tutor exposure throughout the sensitive period or re-exposure to the social tutors during the plastic song stage had higher approaching rate (Fig. 2a), and these birds also had a significant better song imitation compared to the birds were tutored after 45 dph, or tutor song playback (Fig. 2c-d).

Based on our results, we speculate (in revised Discussion, Page 11, Lines 337-346) that juvenile approaching behavior serves an important function to facilitate a juvenile’s instant access and selectively attend to a re-exposed or repeatedly exposed adult tutor and

fine-tune its previously memorized “template” during sensorimotor integration. We have substantially revised the Results and Discussion sections.

- Lines 386-392: was manual scoring used for the dopamine group only or for all experimental birds? Why were not the dopamine group tracked?

Our response: Manual scoring was used for all of our experimental birds, including the dopamine manipulative group (see our revised Methods, Page 15). The birds were first video recorded and then were manually scored by three scorers. We used a tracking software to manually calculate the distance, the duration, and onset timing of the approaching behavior. Automatic tracking was only used for measurement of overall approaching rate (that is, approaching under all social conditions, singing or not, see our earlier reply). As mentioned earlier, because automatic tracking software was not able to detect sounds and identify singing-initiated listening/approaching behavior. We had to use manual scoring to do most of our data analysis.

REVIEWERS' COMMENTS:

Reviewer #2 (Remarks to the Author):

The authors have adequately addressed most of my concerns, at least to the extent that this was possible given that the experiments were already done. I also appreciate the careful answers to the reviewers' questions, the new experiments and new figures. The one thing that I think the editors should insist on is to include why the authors disagree with Ropers /Zann and list the evidence for distinct sensory and sensory-motor phases in zebra finches, since this is central to study's. The authors address this in the answer to reviewers and show also do this in the paper. Just ignoring/omitting a reference that one does not agree with is not the solution I think.

I would also like to clarify that my comment about the (by European standards) small cage size was not in the context of scoring. I did not doubt that scoring was done well. I am arguing that in a cage of those dimensions chance occupancy of perches is different than in a larger cage and thus correlations between bird A and bird B in space and time are higher to occur by chance than in a larger cage. However, since not all behaviors correlated as shown by the additional analyses this is not a point I worry about very much.

Other than these comments I think the manuscript is now definitely a nice addition to the literature.

Reviewer #3 (Remarks to the Author):

The authors have fully addressed my comments, and the revised manuscript is ready for publication.

Reviewer#2

The one thing that I think the editors should insist on is to include why the authors disagree with Ropers /Zann and list the evidence for distinct sensory and sensory-motor phases in zebra finches, since this is central to study's. The authors address this in the answer to reviewers and should also do this in the paper. Just ignoring/omitting a reference that one does not agree with is not the solution I think.

Our response: Because researchers have different views about the extent of overlap between sensory and sensory-motor learning phases in zebra finches, we decided to present both views in Introduction, while citing an review article from Mooney 2009 and adding an article from Liu 2004 paper to show that the development of sensory learning phase precedes the onset of motor learning phase, we also added a short sentence and included the article from Ropers and Zann 2009 to show their view of the complete overlap between these two learning phases.